# Mantle Evolution of Asia Inferred from Pb Isotopic Signatures of Sources for Late Phanerozoic Volcanic Rocks

**Sergei Rasskazov** [1,2,*], **Irina Chuvashova** [1,2], **Tatyana Yasnygina** [1]  and **Elena Saranina** [1]

1   Institute of the Earth's Crust, Siberian Branch of Russian Academy of Sciences, 664033 Irkutsk, Russia; chuvashova@crust.irk.ru (I.C.); ty@crust.irk.ru (T.Y.); e_v_sar@mail.ru (E.S.)
2   Geological Faculty, Irkutsk State University, 664003 Irkutsk, Russia
*   Correspondence: rassk@crust.irk.ru; Tel.: +7-3952-511659

**Abstract:** We present a systematic study of Pb isotope ages obtained from sources of the late Phanerozoic volcanic rocks from unstable Asia and also volcanic rocks and kimberlites from stable regions of the Siberian and Indian paleocontinents. In the mantle sources, we have recorded events of the Early, Middle, and Late epochs of the Earth's evolution. Evidence on the Early epoch are preserved in sources of the protolithosphere and viscous lower protomantle likely generated from the Hadean magma ocean about 4.51 and 4.44 Ga and in sources of the viscous upper mantle that acquired low μ and elevated μ (LOMU and ELMU) signatures in the early Archean (4.0–3.7 Ga). The Middle and Late epochs are denoted by sources of the viscous upper mantle that was generated, respectively, in the late Archean-Paleoproterozoic (2.9–2.6 Ga and 2.0–1.8 Ga) and in the Neoproterozoic-late Phanerozoic (0.7–0.6 Ga and < 0.25 Ga). Our results show the specific role of the mantle beneath unstable Asia in terms of globally varied μ signatures and the same mantle epochs in sources of the late Phanerozoic volcanic rocks and kimberlites from stable regions of the Siberian and Indian paleocontinents, but with high μ (HIMU) signatures that are distributed worldwide and explained by sulfide sequestration of Pb from the mantle to the core. We refer the LOMU-ELMU mantle sources to the Asian high-velocity lower mantle domain and propose that the HIMU generating processes were focused mainly in the South Pacific and African low-velocity lower mantle domains in the Middle Mantle Epoch of the Earth's evolution due to influence of the unbalanced solid core.

**Keywords:** volcanic rocks; late Phanerozoic; Pb isotopes; mantle; HIMU; Asia

## 1. Introduction

### 1.1. Backgrounds

The modern Earth's mantle is globally heterogeneous. High-velocity domains were determined in the lower mantle under Asia and North America, whereas low-velocity ones under the southern Pacific Ocean and Africa [1]. High-resolution seismic tomography [2] shows a significant velocity contrasts in the uppermost (up to 660 km) and lowermost (1800–2900 km) parts of the mantle and a small contrast at its intermediate depths (660–1800 km).

The global Earth's heterogeneity was first indicated by high $^{208}Pb/^{204}Pb$ ratios ($\Delta 8/4 > 60$) in ocean island basalts (OIB) of the 30th degrees south latitude, called the DUPAL (Dupré–Allègre) anomaly [3]. Later, sources of this anomaly were connected with low velocities of seismic waves in the lower mantle [1]. This isotopic signature was defined in basalts from the central part of the high-velocity Asian lower mantle domain as characteristic of the Neoproterozoic Tuva-Mongolian paleomicrocontinent, in contrast to DUPAL-free basalts from the adjacent East Tuva and Dzhida

Caledonian zones [4]. The origin of the DUPAL anomaly as an indicator of the deeper low-velocity mantle domains is uncertain.

From geochemical and geophysical constraints, the South Pacific and African low-velocity mantle domains were connected also with the HIMU extreme values ($^{206}Pb/^{204}Pb > 20$). It was proposed that such values were associated with the deep roots of mantle plumes [5] or with delamination of the carbonatite-metasomatized subcontinental lithospheric mantle [6]. No attempts have been done, however, to define the nature of magmatic sources related to the Asian high-velocity mantle domain in terms of their $^{238}U/^{204}Pb$ ($\mu$) values.

Geodynamic assessments of Asia, as compared with South Pacific, were controversial. The hypothesis of convective rise of a hot material from the core–mantle boundary in Asia [7,8] contradicted the conceptual statement, in which Asia was considered as lacking any signs of plume-type magmatism [9]. Later, low velocities in the lower mantle beneath South Pacific were explained by heating of a slab cemetery related to the supercontinent Rodinia, in contrast to high velocities under Asia that were considered as evidence of a slab storage coeval with the supercontinent Laurasia [10]. This assumption was not consistent with 3–2 Ga age estimates obtained for OIB sources of the South Pacific Isotopic Thermal Anomaly (SOPITA) [11–13]. Besides, it was not confirmed by obtained ages of sources for volcanic rocks from Asia [14].

In multiple papers, sources of oceanic basalts were referred to global endmembers EM1, EM2, HIMU, and DMM (respectively, enriched mantle 1 and 2, high $\mu$, and depleted MORB mantle). OIB source material was assumed to be recycled into the oceanic mantle by 3–2 Ga subduction [13,15]. From analyses of Pb and Os isotopic ratios, the DMM component, denoted MORB, was inferred to be an isolated reservoir relative to other global endmembers and was excluded from the discussion of the OIBs as derivatives of the Bulk Silicate Earth [16,17]. The HIMU compositions we recognized in different regions of the world [5,6,11,18]. The HIMU value ($^{206}Pb/^{204}Pb = 19.6–19.9$, $^{207}Pb/^{204}Pb = 15.60–16.66$) was assigned also to the common mantle component of the Cenozoic anorogenic volcanic rocks of the Mediterranean, Arabian Plate, and adjacent areas [19–21]. Besides, sources of oceanic basalts were designated as derivatives of the primordial mantle in terms of isotopic signatures of noble gases [22,23].

The origin of sources for continental volcanic rocks was often interpreted in terms of the isotopic systematics of oceanic basalts. Such interpretations, however, met contradictions arising from differences in the formation history and existing differences in the dynamics of the mantle under oceans and continents. Under oceans, mantle material has long lost its connection with the accessible Earth structures; under continents, such a connection was often obvious i.e., [24]. Under oceans, lithospheric plates are driven by mantle convection; under continents, mantle convection is limited or absent. The erupted mantle melts in unstable Asia are almost not contaminated with crustal material [14] that opens up the possibility of their use for assessments of isotope-geochemical signatures of the mantle.

The most extensive mobile region of the world, recognized in Asia by high seismicity and present-day block motions of the Earth's surface, is located between stable Eurasia in the north and India and Arabia in the south. Stable cores, the Siberian and Indian paleocontinents, are assumed to behave rigidly [25–31]. The Siberian paleocontinent was integrated into the geological structure of Eurasia in the early Paleozoic, although the paleooceans (Solonker, Turkestan, Mongolia-Okhotsk) closed to the south of it in the Late Paleozoic and partially in the Mesozoic [27]. The Indian paleocontinent, drifted from the south to north and collided with Asia 65–32 Ma ago [32,33].

The global age distribution of kimberlite magmatism suggests that this kind of a mantle process was more prominent after 1.2 Ga (notably between 250–50 Ma), than before 2 Ga ago (i.e., in the Paleoproterozoic and Archean). An incipient melting regime that results in generation of kimberlite magmas likely requires advanced conditions of the maturing continents [34]. Taking into account inconsistency between Pb- and Sr–Nd–Hf-isotopic data, due to the contrast effects of Pb concentration in sulfides and Sr–Nd–Hf in silicates [17,18,35–37], we have identified sources of Asian volcanic rocks in the framework of Pb–Pb isotope systematics as the basic approach to discrimination of the whole data set in terms of age estimates for protoliths in source regions [14]. The aim of this paper is to

present Pb isotope ages of sources for recent volcanic rocks of unstable Asia in comparison with ages of sources for the older traps and kimberlites in stable regions of the Indian and Siberian paleocontinents in the global geodynamic context. The methods used for analytical determinations of Pb isotope ratios are presented by Harris [38] and Rasskazov et al. [4,39,40].

### 1.2. Isotopic Age Assessments of Mantle Sources for Volcanic Rocks

Common generation of source material assumes isochronous relationships between its constituencies. Incubation timing is estimated from distribution of data points on a diagram of uranogenic Pb isotope ratios ($^{207}$Pb/$^{204}$Pb versus $^{206}$Pb/$^{204}$Pb). A compact isometric data field indicates either the modern homogenization of a source material or $\mu$ = const in an older source. In any case, a compact isometric set of data points erases age information. Complete isotopic homogenization of a source, providing a U–Pb isotope system with an initial Pb isotope ratio, can be achieved by convective mixing of melted material. Differentiation of a source by $^{238}$U/$^{204}$Pb ($\mu$) values results in time-integrated accumulation of uranogenic isotopes $^{206}$Pb and $^{207}$Pb.

A linearity of data points is due to the different half-lives of the parent isotopes $^{238}$U and $^{235}$U that are kinetically inseparable in any geological media. In Late Cenozoic mantle-derived volcanic rocks, $\mu$ values are low; therefore, a time-integrated shift of their Pb isotope compositions is negligible. Age calculation of a source region for volcanic rocks requires only a series of uranogenic Pb data points without measurements of U and Pb abundances. To estimate a source age of older volcanic rocks, however, additional information on U and Pb concentrations in each sample with calculations of the initial Pb isotope ratios is required. An age obtained, adjusted to the present, is summed up from the time interval of closed incubation of the viscous mantle material and the time passed since the volcanic eruption.

Unlike $^{207}$Pb/$^{204}$Pb—$^{206}$Pb/$^{204}$Pb isochrons, those in the Sm–Nd, Lu–Hf, and Rb–Sr isotope systems are obtained from ratios of both parent and daughter radionuclides and chemical element concentrations. The latter depends on degree of partial melting in a source region; therefore, an isochron for source material must be corrected using mineral-melt distribution coefficients. This brings uncertainty to age assessment.

### 1.3. Previous Interpretations of Isotopic Data

Isotope-geochemical studies of volcanic rocks show different mantle components with particular interpretations in each region of Asia [4,14,40–56].

In terms of Pb, Nd, and Sr isotope ratios, potassic lavas from the Wudalianchi, Erkeshan, and Keluo volcanic fields (WEK zone) were interpreted as the EM1 OIB end-member of the subcontinental lithospheric mantle in contrast to potassic-sodic basalts from other areas in East China that were attributed the DMM OIB end member presumably derived from the asthenospheric mantle [55,57,58]. The asthenosphere was assumed to invade the lithospheric mantle in a subduction zone associated with Japan's islands. Decompression melting of the lithospheric mantle was examined as the leading process that was complicated by melting of the subducted oceanic crust.

After definitions of the FOZO (Focal Zone) and C (Common) components from convergent trends of data points on diagrams of different isotope ratios of oceanic basalts [16,59], the same approach was implemented for identifications of common components in continental basalts from different regions. In East Asia, a common depleted sub-lithospheric component was proposed from interpreting Nd, Sr, and Pb isotope ratios of Late Cenozoic alkaline basalts erupted on the eastern continental margin of China [56]. Lead depletion, niobium and tantalum enrichment, high Ce/Pb, Nb/U ratios, and low La/Nb ratios in these rocks were examined as specific trace-element signatures of ocean island basalts. In this systematics, the common component of the depleted sub-lithospheric mantle of the region was designated by lavas from the Tashan (16.3 Ma), Nushan (0.72–0.55 Ma), and Fangshan (9.1–9.4 Ma) volcanoes. In basalts from Northeast and Southeast China, this common sub-lithospheric component was assumed to be mixed with the enriched lithospheric mantle material similar to the EM1 OIB (low

$^{206}$Pb/$^{204}$Pb) and with the enriched lithospheric mantle material similar to the EM2 OIB (with a high $^{206}$Pb/$^{204}$Pb), respectively. The EM1 OIB component was attributed to rocks from the Wudalianchi volcanic field (<2.5 Ma) and the EM2 OIB one to those from the Niutuo volcano (17.9–16.7 Ma).

Other isotopic systematics of volcanic rocks from Asia assumed different relations between components of oceanic basalts. For instance, Pb, Nd, and Sr isotope ratios of volcanic rocks from Southeast Asia were interpreted within the framework of three hypothetical mixing models of endmembers. The first model yielded a mixing line of depleted mantle (DM) and the mantle with high $^{238}$U/$^{204}$Pb (HIMU) (it was equivalent to the NHRL, Northern Hemisphere Reference Line). The endmembers of the second model were the enriched component EM1 and the hybrid component N-MORB (normal mid-ocean ridge basalt), resulted from the DMM and HIMU mixing. The mixing line of model 2 was one of the components of the third model, which assumes the EM2 as another component. These systematics emphasized the difference between Pb isotope signatures of East Asian basalts and the Pacific MORB and their similarities with the Indian OIB [60,61].

From Sr, Nd, Pb, and Hf isotope ratios, sources of volcanic rocks in East Asia were subdivided into two domains. One of these (Sin-Meng) was referred to the asthenosphere, composed of a mixture between the DMM OIB and EM1 OIB endmembers, another one (South China) to the asthenosphere, composed of a mixture between the DMM OIB and EM2 OIB end-members [43]. One more interpretation assumed mixing between the FOZO OIB and LoMu (low μ) endmembers; the latter was referred to the continental lithosphere [42].

Study of volcanic rocks of the last 37 Ma in East Asia showed an enriched signature of the common component: $^{87}$Sr/$^{86}$Sr = 0.7052, $^{206}$Pb/$^{204}$Pb = 17.55, $^{207}$Pb/$^{204}$Pb = 15.52, $^{208}$Pb/$^{204}$Pb = 37.76. This signature was obtained for volcanic rocks erupted in the back-arc side of the Northeast Honshu arc between 30 and 20 Ma and for volcanic rocks of the southern part of the Transbaikal melting domain. Volcanic rocks with this common component preceded Late Cenozoic subduction of the Pacific Slab that started in the Northeast Honshu arc about 18 Ma ago. Melts with depleted slab-derived isotopic signatures erupted in the back-arc side region between 15 and 6 Ma. The role of more depleted melts increased in the last 4.4 Ma [39,62,63].

In Inner Asia, late Cenozoic volcanic rocks from Eastern Sayans showed a depleted signature of a common component: $^{87}$Sr/$^{86}$Sr = 0.7041, $\in$ Nd = +3, $^{206}$Pb/$^{204}$Pb = 18.1–18.2, $^{207}$Pb/$^{204}$Pb = 15.53, $^{208}$Pb/$^{204}$Pb = 38.2. A similar common component was identified in volcanic rocks from the western part of the Vitim volcanic field but was not detected in its eastern part [4,14].

Age assessments of volcanic rock sources are disputable. The first obtained Pb isotope data set of volcanic rocks from the Changbaishan (Changbai), Hannuoba, Mingxi, Datong, Kuandian, and Wudalianchi areas was approximated by the NHRL with a slope that corresponds to an age of 1.77 Ga [57]. Later, Pb isotope data of volcanic rocks and their deep inclusions in the Kuandian and Hannuoba fields were used to designate episodes of metasomatic enrichment of the continental lithosphere about 3.38 and 2.65 Ga [50]. Besides, wide intervals of Pb isotope ratios were defined in the Linju Miocene basalts (Shandong Peninsula) that overlapped the whole ranges of these ratios in volcanic rocks from East China. Data points, distributed along a single line with a slope of about 2.57 Ga, were referred to a secondary isochron of a material melted from an old lithospheric mantle keel [52].

For a source region of Wudalianchi potassic liquids, an age of about 1.5 Ga, estimated from a model of ore leads modified after Stacey and Kramers [64], was interpreted assuming their origin from a slab stagnated in the mantle transition layer [44]. Used in these estimates were the Pb isotope data on volcanic rocks from historical eruptions of the Laoheishan and Huoshaoshan volcanoes. The same idea about the origin of potassic volcanic rocks of the Wudalianchi volcanic zone due to melting of a stagnated slab was presented with another model Pb isotope age estimate of 2.2 Ga [65]. More detail trace element and Pb isotope study of volcanic rocks from the Wudalianchi volcanic field revealed, lateral heterogeneity of mantle sources beneath different volcanoes. The obtained Pb–Pb isotope ages varied from 1.88 Ga (Laoshantou and Old Gelaqiushan flows) to the Quaternary (Molabushan cone) [14,40].

In Central Mongolia, upper Cretaceous-Paleogene volcanic rocks from Gobi and Neogene-Quaternary volcanic rocks from Hangay yielded contrast Pb isotope ages of sources. The former showed an array with a slope close to the meteorite geochron, while the latter indicated another array with a slope of about 0.66 Ga [14].

### 1.4. Strategy for Analysis of Pb Isotope Data

In unstable Asia, occurring between stable regions of the Indian and Siberian paleocontinents, melting anomalies are distributed along the Japan-Baikal geodynamic corridor and Tibet-Indochina geodynamic belt. The former is designated by the Baikal, Circum-Ordos, and East China rift systems and Central Asian and Olekma-Stanovoy orogenic systems, the letter by the Indo-Asian zone of collision and its Indochina flank (Figure 1).

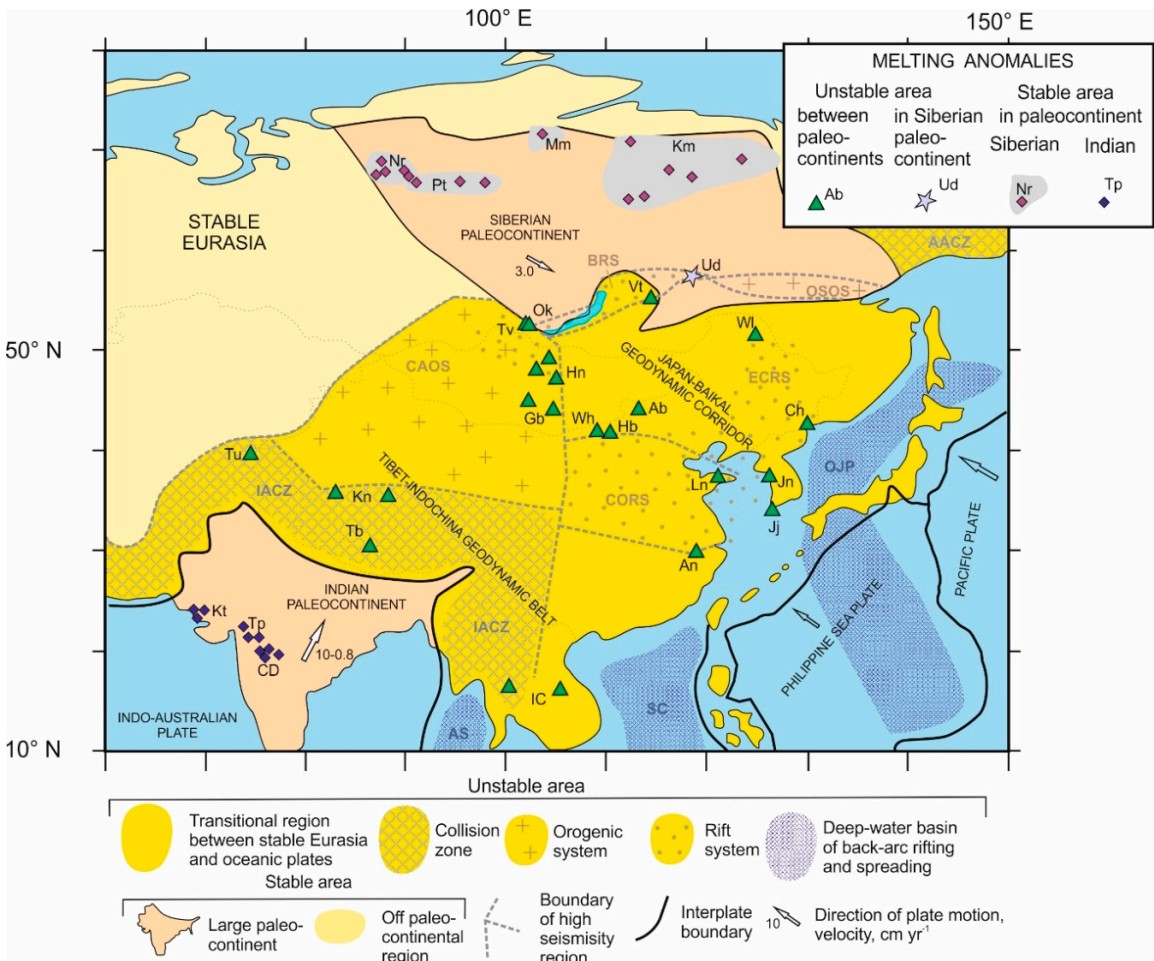

**Figure 1.** Distribution of selected late Phanerozoic melting anomalies in unstable and stable parts of Asia. The Japan-Baikal geodynamic corridor: Ch—Changbai, Jn—Jeungok, Wl—Wudalianchi, Ab—Abaga-Dalenyuor, Ln—Linju, An—Anhui, Hb—Hannuoba, Wh—Wulanhada, Gb—Gobi, Hn—Hangay, Vt—Vitim, Ud—Udokan, Tv—Tuva, Ok—Oka. The Tibet-Indochina geodynamic belt: Tu—Tuyun, Kn—Kunlun, Tb—Tibet, Ic—Indochina. The Indian paleocontinent: Kt—Kutch, Tp—Tapi Rift, CD—Central Deccan. The Siberian Paleocontinent: Nr—Noril'sk, Pt—Putorana, Mm—Meimecha, Km—Siberian kimberlite province. Mobile systems of unstable Asia: Indo-Asian (IACZ) and American-Asian (AACZ) collision zones; Central Asian (CAOS) and Olekma-Stanovoy (OSOS) orogenic systems; Baikal (BRS), Circum-Ordoss (CORS), and East China (ECRS) rift systems; Okhotsk-Japan-Philippine (OJP), South China (SC), and Andaman Sea (AS) deep-water basins of back-arc rifting and spreading. The mobile systems and Japan-Baikal geodynamic corridor are shown after [14,66].

Volcanism of unstable Asia marks deep processes of the latest geodynamic stage that evolved to the present-day unstable geodynamic state during the last 90 Ma. Previous Early Cretaceous volcanism did not fit into this evolution [14]. The stability of the Indian and Siberian paleocontinents is shown by low seismicity and heat flow, as well as by lack of block motions and Late Cenozoic volcanism. In the Indian paleocontinent, traps and kimberlites of the Deccan large igneous province (LIP), dated within the age interval of 67–60 Ma [67,68], are coeval to an early phase of the latest geodynamic stage in unstable Asia. In the Siberian paleocontinent, the latest magmatism is marked by kimberlite pipes, some of which are as young as 45 Ma and, therefore, also partially overlap the latest geodynamic stage. Traps of the Siberian LIP are dated back to the Permian-Triassic boundary [39,69,70] and references therein.

The southeastern part of the Siberian paleocontinent was unstable due to involvement in the rifting of the Baikal mobile system and orogenesis of the Olekma-Stanovoy mobile system. In the rift system, lavas of the Udokan volcanic field erupted in the last 14 Ma, in the orogenic system, lavas of the Tokinsky Stanovik volcanic field erupted in the 0.6–0.3 Ma time interval. The latest tectonic reactivation of these areas was preceded by the Late Phanerozoic tectonic motions accompanied by Late Paleozoic granite and Mesozoic basic potassic magmatism [39,71] and references therein.

Since the modern geodynamics of the stable and unstable parts of Asia differs, the strategy of this study implies a comparative study of sources for the Late Phanerozoic mantle-derived magmatism, on the one hand, as an indicator of similarities and differences between deep processes of stable and unstable areas, and, on the other hand, as an indicator of peculiarities in the development of the reactivated southeastern part of the Siberian paleocontinent.

Coverage of the entire Earth's history by ages of source regions for volcanism in unstable Asia [14] implies the possibility of tracing the evolution of the mantle from its primordial state to the present similar to the common (galena) leads evolution [64]. The B-type lead in the Amitsoq gneisses (Isua Belt, south of West Greenland) was approximated by a curve with $\mu_2 = 9.7$, starting from 3.7 Ga. This moment in the Earth's history was taken as the peak of the crust-forming event. The preceding processes were described by the accumulation curve of radiogenic leads at $\mu_1 = 7.2$. The model was used in practice for rough age estimates of "conformable" leads in ore deposits. Some anomalous (J-type) ore leads, however, yielded data points to the right of the geochron that did not match the selected parameters of the Stacey and Kramers model [72].

In the early Earth evolution $\mu$ values changed due to loss of volatile Pb (relative to refractory U) during meteorite bombardment, similar to the loss by Proto-Earth of other volatile elements such as Zn, He, Ne, etc. [35,73–75]. The content of moderately volatile Pb in the mantle may be also governed by sulfide segregation into the core and only slightly affected by late veneer [76].

Unlike the Stacey and Kramers model and similar models based on the assumption of a variable $\mu$ value over time i.e., [77], the model of Rasskazov et al. [39] implied no preliminary assumption of $\mu$ values. Calculations were performed assuming a constant $\mu$ value derived from conjugated calculations of (1) the accumulation of radiogenic Pb according to the Holmes–Houtermans model and (2) the evolution of Pb along diffusion discordia according to the Wasserburg model. The conjugated model was based on the "hot Earth" hypothesis. Radiogenic Pb was first accumulated in the molten Earth with shift from the meteorite Geochron along the Concordia and then in an external viscous shell of the Earth with a shift from the Concordia along the diffusion discordia. Respectively, the conjugated calculations determined (1) a moment $T$ of transition from convecting to a viscous state of medium with the closure of the U–Pb isotope system and (2) a moment $t$ of Pb separating from U into an ore mineral. The conjugated model was applied for definitions of different scenarios of the ore lead generations in the early Precambrian crust of the Gargan block and the southern margin of the Siberian paleocontinent. In the former, B-type leads were separated from the Hadean protolith ($T = 4.31$ Ga) in the Paleo-Mesoproterozoic time interval ($t = 2.3$–1.4 Ga); in the latter, J-type leads were evolved from the early Archean protolith ($T = 3.82$ Ga) in the Paleoproterozoic-Phanerozoic time interval

($t$ = 1.80–0.25 Ga). Respectively, the μ value in the younger protolith (20.1) was sufficiently higher than in the older one (11.0).

Similarly, a basic assumption on transition from the molten Earth (planetary magma ocean) to the viscous layered mantle is reasonable for consideration of Pb isotope evolution in source regions of recent melting anomalies with a variable μ, reflected in the Pb isotope evolution of the early Earth. In the Earth model of variable viscosity, adopted here for age systematics of continental volcanic rocks, Pb isotope compositions are referred to as bulk composition of a viscous protomantle generated due to cooling of the planetary magma ocean.

In the Earth's evolution models, the mantle viscosity increased downwards by two orders of magnitude [78]. From the depth of 660 km, the viscosity increases sharply, reaching its maximum at 1000 km [79]. The lower mantle is considered as a part of the main convective system, but its high viscosity could actually lead to long-term isolation of large lower mantle blobs from the rapidly convecting upper mantle [80]. Therefore, the global evolution of the Earth is explained in the framework of a hypothesis of two-layer mantle convection that suggests convective instability of the upper mantle and the preservation of the BSE material in the lower mantle. It is suggested that the Rayleigh number changed in time with the transition from Precambrian convection, not accompanied by slab subduction into the lower mantle, to the modern global structure of the Earth, characterized by oceanic slab penetration into the lower mantle [81,82].

The diagrams of uranogenic Pb isotope ratios (Figures 2 and 3) indicate a locus that corresponds to a transition from the magma ocean of the molten Earth to a Low μ Viscous Protomantle (LOMUVIPMA), starting from the primordial lead of the Canyon Diablo meteorite and passing to the right of the meteorite geochron with a slope of 0.625034. A slope of the LOMUVIPMA geochron (0.600827), corresponding to an age of about 4.51 Ga, and a range of a Low μ Viscous Protomantle Reservoir (LOMUVIPMAR) are derived from Pb isotope compositions of volcanic rocks from the northwestern part of the Udokan volcanic field.

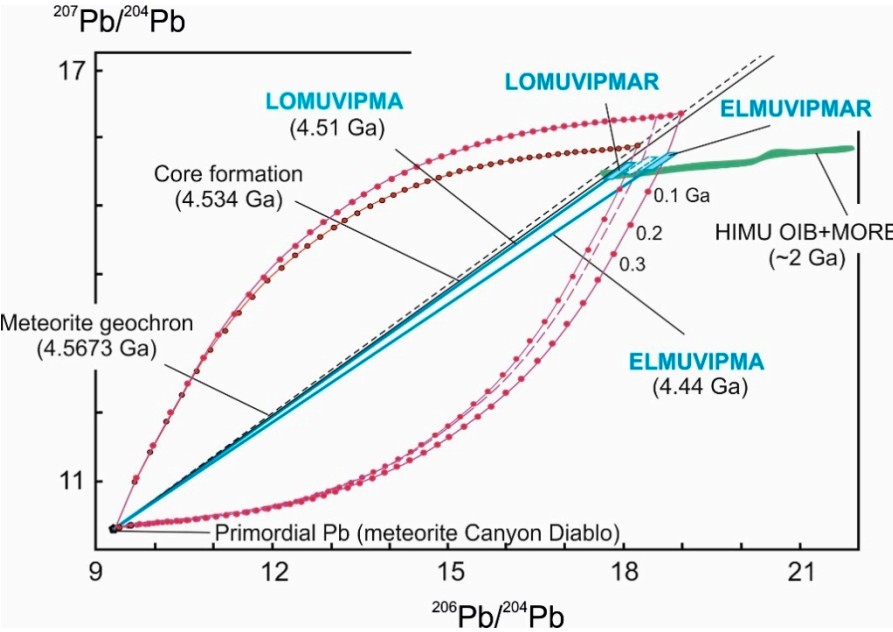

**Figure 2.** Schematic diagram of uranogenic Pb evolution. Shown are the meteorite geochron [83], the core formation line, the HIMU OIB + MORB trend [18], and the LOMUVIPMA and ELMUVIPMA lines (this work). The LOMUVIPMA and ELMUVIPMA lines yield limited compositions LOMUVIPMAR and ELMUVIPMAR that originated from stepwise solidification of the planetary magma ocean. All geochrons are focused on the primordial lead isotope ratios determined in a troilite from the Canyon Diablo meteorite (Nanton): $^{207}$Pb/$^{204}$Pb = 10.307094, $^{206}$Pb/$^{204}$Pb = 9.305875 [84].

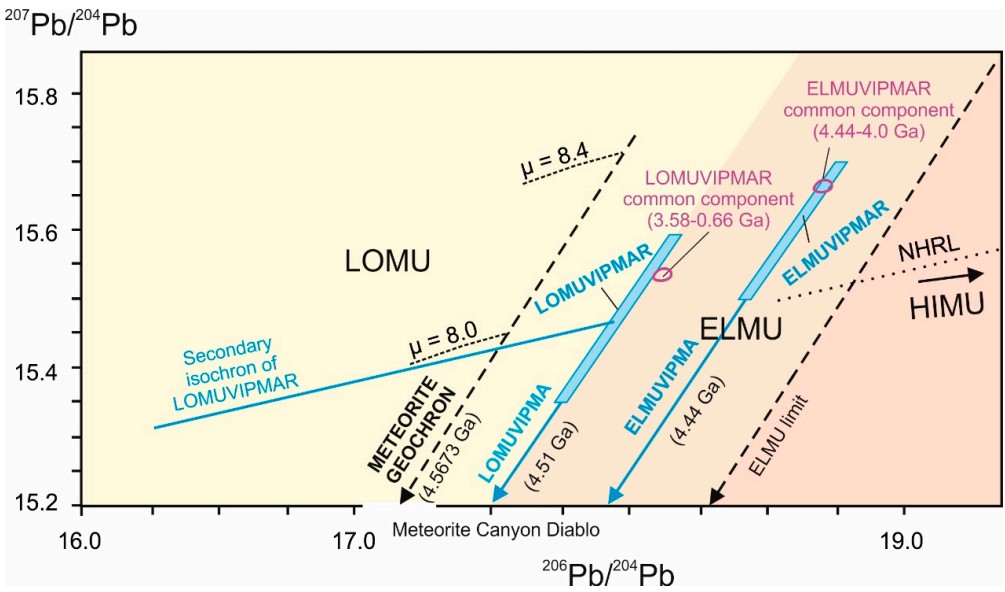

**Figure 3.** General Pb isotope discrimination of mantle sources for volcanic rocks of Asia on a plot of uranogenic Pb isotope ratios. The empirical geochrons LOMUVIPMA and ELMUVIPMA with slopes corresponding to the ages of about 4.51 and 4.44 Ga are shifted to the right of the geochron, the slope of which corresponds to the age 4.5673 Ga of the highest temperature Ca–Al inclusions of meteorites [83]. Discriminated are the low μ, elevated μ, and high μ (respectively, LOMU, ELMU, and HIMU) compositions. The LOMUVIPMAR and ELMUVIPMAR are determined by ranges of volcanic rocks, derived from the Udokan 1 source and Kunlun–Abaga-Dalenuor sources, respectively. The common components ELMUVIPMAR and LOMUVIPMAR are defined for volcanic rocks of the Kunlun and Tuva 1—Hangay sources, respectively.

Due to irregular cooling of the convective system in the Earth's magma ocean, its residual portion of convective material yielded an additional geochron of Elevated μ Viscous Protomantle (ELMUVIPMA). A slope of the ELMUVIPMA geochron (0.572614), corresponding to an age of about 4.44 Ga, and a range of an Elevated μ Viscous Protomantle Reservoir (ELMUVIPMAR) are obtained from Pb isotope ratios of volcanic rocks of the Kunlun and Abaga-Dalenuor volcanic areas.

We suggest that LOMUVIPMAR and ELMUVIPMAR may represent primary viscous mantle materials that produced secondary compositions in the Earth's history. Some secondary isochrons diverged from LOMUVIPMAR towards a low $^{238}$U/$^{204}$Pb (LOMU) part of the Pb–Pb isotope diagram. Others are located around ELMUVIPMAR and referred to Elevated μ (ELMU) compositions.

In the presentation of results, we first define a position of viscous protomantle loci and then designate ages of the subsequent mantle evolution in the stable and unstable parts of the continent. For volcanic rocks of different areas, we also define common components (Figure 3).

## 2. Results

### 2.1. Age Sequence of Protoliths in Mantle Sources

The obtained ages of protoliths in mantle sources of volcanic rocks erupted in unstable Asia, are presented on a time scale as a sequence of six time intervals. The first interval (Hadean, 4.51–4.44 Ga ago) indicates a generation of viscous protomantle reservoirs from the planetary magma ocean and an early processing of their protoliths. Later intervals are: II—early Archean (4.0–3.6 Ga ago), III—late Archean (2.9–2.6 Ga ago), IV—Paleoproterozoic (2.0–1.8 Ga ago), V—Neoproterozoic (about 0.7–0.6 Ga ago), and VI—late Phanerozoic (<0.25 Ga ago). The intervals obtained for sources of volcanic rocks from unstable Asia are compared with ages of sources for volcanic rocks from stable paleocontinents (Figure 4).

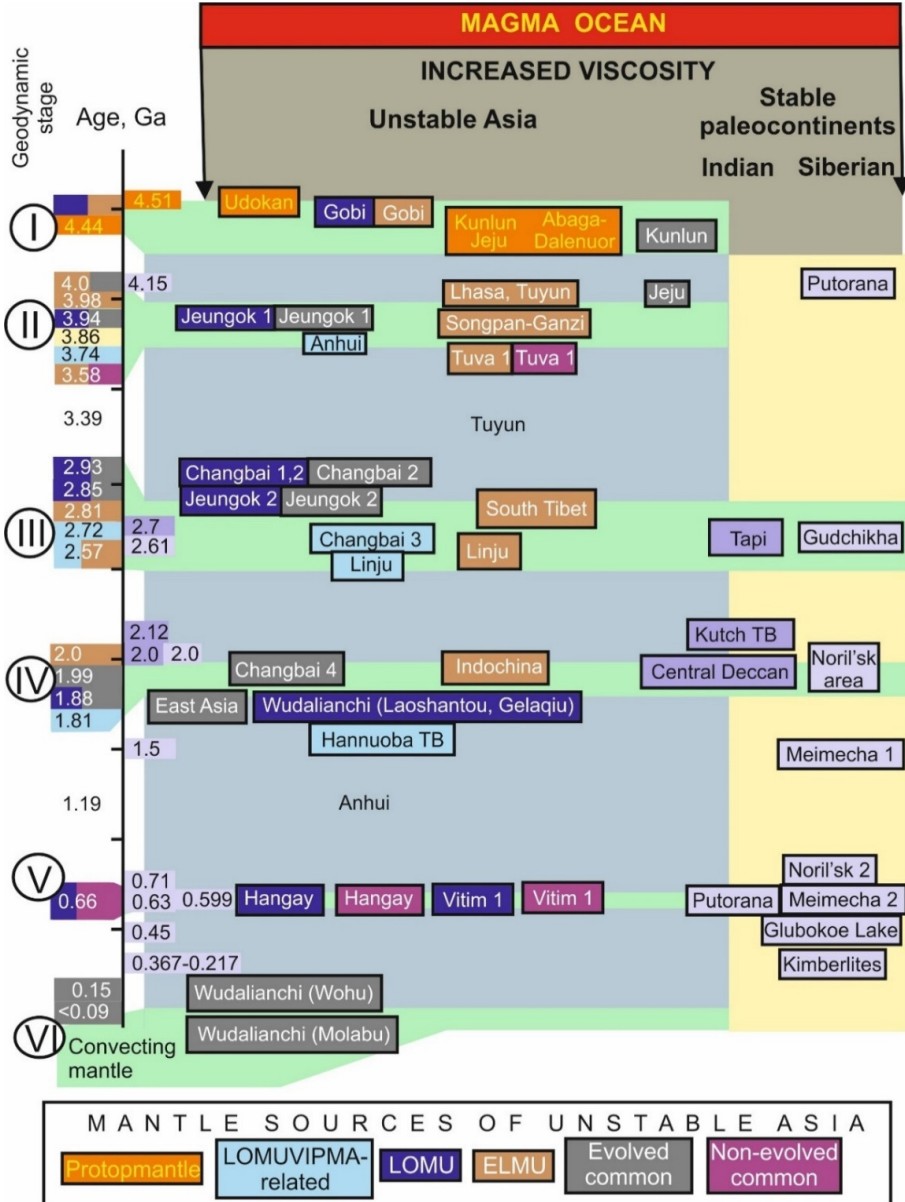

**Figure 4.** Correlation of Pb isotope ages of sources for late Phanerozoic melting anomalies in the viscous mantle of unstable Asia and Indian and Siberian paleocontinents. The viscous mantle sources of unstable Asia are divided in terms of the primary origin of the viscous mantle from the magmatic ocean (protomantle, LOMUVIPMAR-related), μ signatures (LOMU, ELMU), and the nature of common components (evolved, non-evolved). Roman numerals in circles indicate the six time intervals of the viscous mantle generation in unstable Asia. No Pb isotope ages were determined in cases of isometric data-point distributions, so such units as the Hannuoba TEPB (transitional, evolved, primitive basalts), Kutch AB (alkali basalts), Wulanhada, Oka, Vitim 2, Vitim 3, and Xiaoguli are not shown on the chart.

## 2.2. Hadean (4.51–4.44 Ga)

The LOMUVIPMA geochron with a slope of about 4.51 Ga symmetrically approximates a scattered set of data points of volcanic rocks from the northwestern part of the Udokan volcanic field designated as the Udokan 1 unit (Figure 5a). A reason to refer this line to a geochron is its orientation at the primordial lead composition of the meteorite Canyon Diablo. Respectively, the LOMUVIPMAR composition is determined by a range of Pb isotope ratios of the Udokan 1 volcanic rocks ($^{207}Pb/^{204}Pb$ from 15.35 to 15.60 and $^{206}Pb/^{204}Pb$ from 17.7 to 18.2).

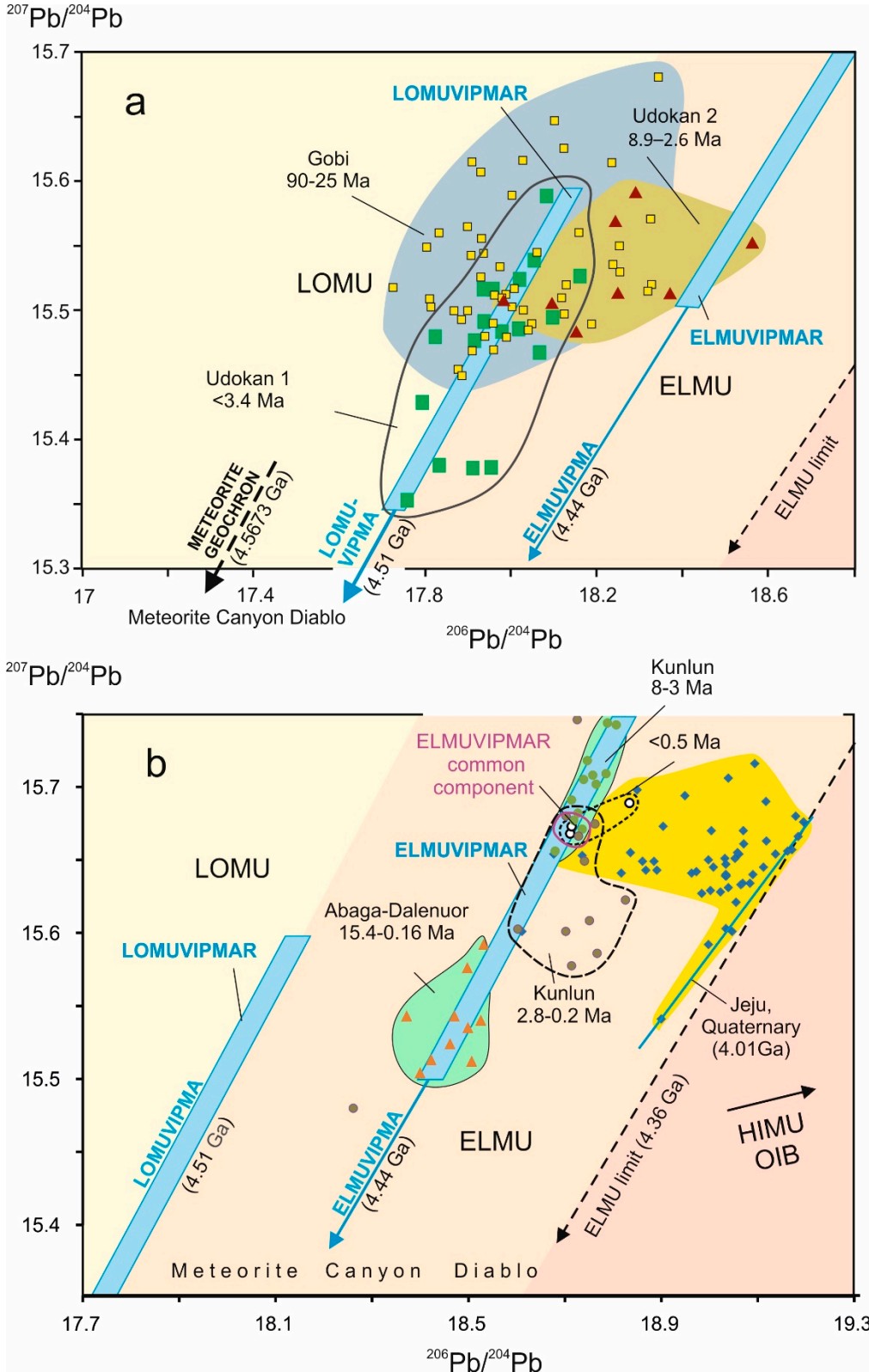

**Figure 5.** Diagrams of $^{207}Pb/^{204}Pb$ versus $^{206}Pb/^{204}Pb$ for identifying Low μ Viscous Protomantle Reservoir (LOMUVIPMAR) (**a**) and Elevated μ Viscous Protomantle Reservoir (ELMUVIPMAR) (**b**). Here and below, a time interval of volcanic eruptions is shown in Ma and a calculated source age of the viscous mantle is designated in Ga with brackets. In diagram (**a**), the LOMUVIPMAR composition is

defined by a data set obtained for volcanic rocks from the northwest of the Udokan volcanic field. In diagram (**b**), the ELMUVIPMAR composition is indicated from combined sets of volcanic rocks of the Kunlun and Abaga-Dalenuor volcanic areas, the ELMU limit of the early Earth is obtained from a maximal $^{206}$Pb/$^{20}$4Pb ration of a data set of Jeju Island volcanic rocks. The common component $^{207}$Pb/$^{204}$Pb = 15.67, $^{206}$Pb/$^{204}$Pb = 18.71 is derived from converging trends of Kunlun volcanic rocks. The same Pb isotope composition is identified in Jeju volcanic rocks. Empirical LOMUVIPMA and ELMUVIPMA geochrons with slopes that correspond to the ages of 4.51 and 4.44 Ga are shifted to the right of the meteorite geochron [83]. Data are from [14,38,42,46,49,85,86].

A more dispersed set of Gobi volcanic rocks is shifted relative to the Udokan 1 set along the geochron line with increasing and decreasing μ. The geochron line is partly overlain by both the Udokan and Gobi data points that may indicate superimposed processes and implies the definition of the Gobi array as the oldest secondary errorchron. Because of the wide scattering of data points, timing of these superimposed processes is not determined.

The ELMUVIPMA locus is obtained for basalts, alkaline basalts, and basanites erupted in the Kunlun volcanic area in the last 18 Ma and also for volcanic rocks of the same compositions erupted in the Abaga-Dalenuor volcanic field in the last 15.4 Ma. The ELMUVIPMAR composition is constrained as a composite range of Kunlun and Abaga-Dalenuor volcanic rocks ($^{207}$Pb/$^{204}$Pb from 15.5 to 15.75 and $^{206}$Pb/$^{204}$Pb from 18.4 to 18.8) (Figure 5b).

Distributed along the ELMUVIPMA line are the data points of the 8–3 Ma unit of volcanic rocks from the Kunlun area. This data field is partially overlapped by the 2.8–0.2 Ma unit advanced to the lower $^{207}$Pb/$^{204}$Pb values. Data points of the last 0.5 Ma volcanic rocks fall into the overlapping field and shift with relative increase of both $^{207}$Pb/$^{204}$Pb and $^{206}$Pb/$^{204}$Pb ratios. The overlapping data field designates the ELMUVIPMAR common component.

A composition similar to the common component of Kunlun volcanic rocks is defined as a non-radiogenic endmember in Quaternary volcanic rocks from Jeju Island, located in the southern part of the Sea of Japan. These volcanic rocks are transitional-tholeiitic basalts and alkaline basalt–hawaiite–mugearite series. Transitional basalts show relatively low $^{207}$Pb/$^{204}$Pb ratios (15.592 and 15.541), in contrast to elevated values in alkaline and tholeiitic basalts (15.601–15.69). Scattering data points have a distinct limit by a line of transitional basalts with a slope 4.01 Ga. This limit designates the highest ELMU compositions in volcanic rocks from Asia. On the whole, the Jeju volcanic rocks may represent a mixing trend between the common ELMUVIPMAR component and viscous mantle material processed about 4 Ga ago.

Plotted in Figure 5a is also the data set of volcanic rocks from the southeastern part of the Udokan volcanic field designated as the Udokan 2 unit. These data set is scattered between the LOMUVIPMAR and ELMUVIPMAR compositions (Figure 5a).

### 2.3. Early Archean (4.0–3.6 Ga)

The early Archean interval is constrained by the ages of volcanic rocks from LOMU (Anhui, Jeungok) and ELMU (Songpan-Ganzi, Lhasa, Tuyun, and Tuva) sources. In the northern part of the Siberian paleocontinent, this interval is correlated with the ages of sources for Putorana traps. No early Archean ages have been obtained for sources of traps from the Indian paleocontinent.

For volcanic rocks from Anhui Province (East China), erupted in the Tanlu fault zone near the northern edge of the Yangtze block all through the Cenozoic, a trend with low $^{207}$Pb/$^{204}$Pb is indicated in the LOMU field to the left and below the LOMUVIPMAR composition. At the beginning of the trend, there is a series of data points of Quaternary volcanic rocks from the Nushan volcano. Data points are distributed along a line with a slope 3.74 ± 0.29 Ga (MSWD = 2.9). The low limit of the data field may denote a mantle processing age of about 1.2 Ga (Figure 6a,b).

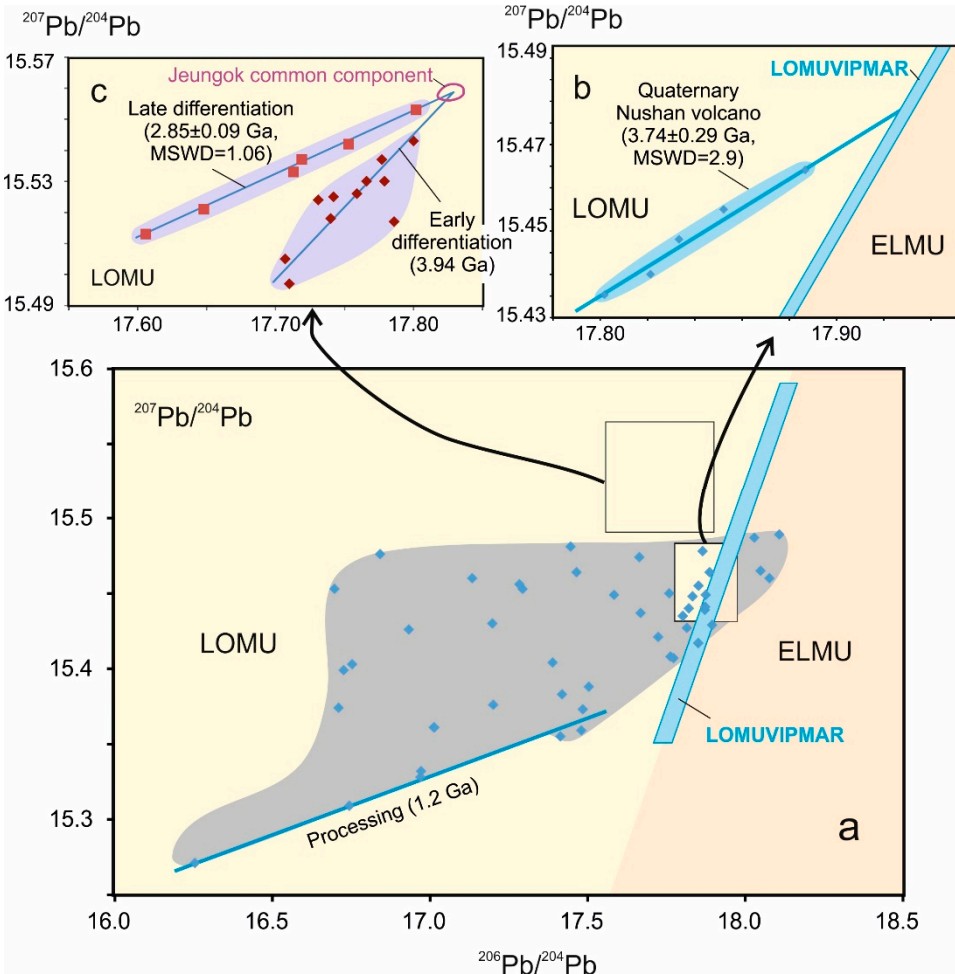

**Figure 6.** Diagrams of $^{207}Pb/^{204}Pb$ versus $^{206}Pb/^{204}Pb$ for volcanic rocks from the Anhui province (**a**), Quaternary volcano Nushan of this province (**b**), and Jeungok volcanic field (**c**). In diagrams (**a**,**b**), the LOMUVIPMAR composition and LOMU–ELMU discriminating line are shown as in Figure 5. In diagrams (**b**,**c**), calculations of the secondary errochron 3.74 Ga and secondary isochron 2.85 Ga, respectively, were performed using the Isoplot program [87]. The Jeungok common component shows relatively radiogenic Pb. Data are from [51,56,88,89].

Another LOMU source pattern is shown by basalts from the Jeungok volcanic field located in the Choogaryong Rift Valley (Korean Peninsula), accommodated at the collisional border between the North China and South China Blocks [51]. Data points are subdivided into 2 arrays with slopes corresponding to ages about 3.94 Ga and 2.86 Ga. The arrays are mutually converged at a point $^{206}Pb/^{204}Pb = 17.84$, $^{207}Pb/^{204}Pb = 15.56$ that denotes a common source material first differentiated about 3.94 Ga ago and again differentiated about 2.85 Ga ago (Figure 6c).

Basanites from the Songpan-Ganzi terrane (Tibet), erupted in the Mid-Miocene through Quaternary, overlap the ELMUVIPMAR composition and show an array with a slope of about 3.86 Ga. Volcanic rocks of other areas in the Indo-Asian collision zone are shifted to the LOMUVIPMAR composition (Figure 7a,b). Adakites from the Lhasa terrane (South Tibet), erupted between 26 and 12 Ma, are approximated by a line with a slope that corresponds to an age of about 3.98 Ga. Although the adakites have crustal geochemical signatures, the age of their source is comparable to those of other early Archean sources of volcanic rocks in South Asia. The 40–38 Ma high-Mg unaltered rocks from South Tibet yield array with a gentler slope that corresponds to an age of about 2.81 Ga.

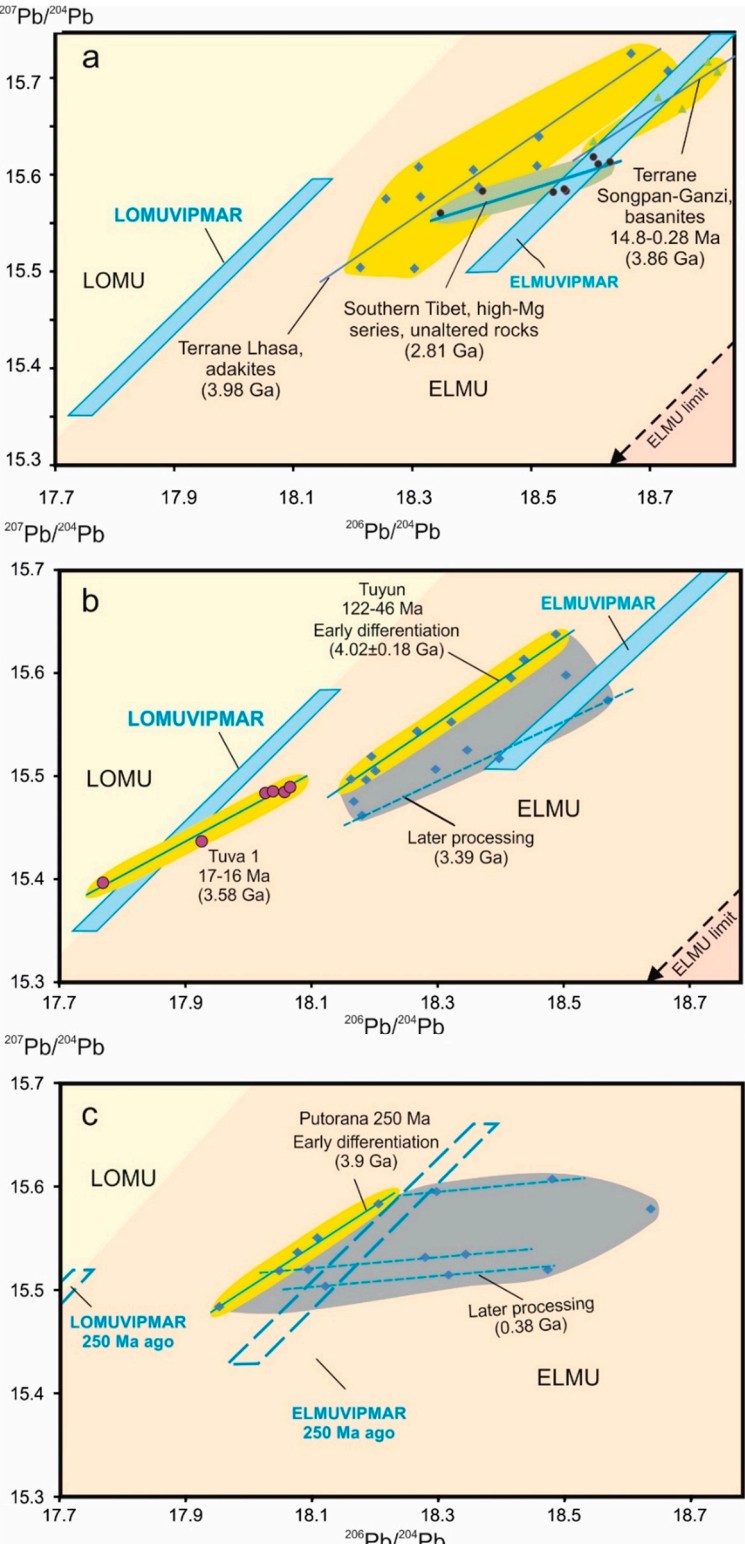

**Figure 7.** Diagrams of $^{207}Pb/^{204}Pb$ versus $^{206}Pb/^{204}Pb$ for volcanic rocks from the volcanic fields of Tibetan Plateau (**a**), Tuyun basin (Chinese Tien Shan) and Tuva (Eastern Sayans) (**b**), and Putorana Plateau (north of the Siberian paleocontinent) (**c**). In diagrams (**a**,**b**), the LOMUVIPMAR and ELMUVIPMAR compositions, as well as LOMU–ELMU and ELMU limit discriminate lines are shown as in Figure 5. In diagram (**c**), data points of traps and the LOMUVIPMAR and ELMUVIPMAR compositions were recalculated to the age of 250 Ma. In diagram (**b**), calculation of the secondary errorchron 4.02 Ga was performed using the Isoplot program [87]. Data are from [53,90–95]. Late Archean (2.9–2.6 Ga).

A 122–46 Ma volcanic series picrobasalt–trachybasalt–basanite–phonothephrite–phonolite, erupted in Tuyuon Basin (Chinese Tien Shan), falls between the LOMUVIPMAR and ELMUVIPMAR compositions. The trend has a slope corresponding to an age of 4.02 ± 0.18 Ga. Some points, plotted below this trend, reflect later processing of a source protolith.

The lower limit of data points corresponds to an age of about 3.39 Ga. Basalts and andesitic basalts, erupted 17–16 Ma ago in the Ulug-Arga range (Eastern Tuva), yield a secondary isochron with a slope corresponding to the youngest age estimate (about 3.58 Ga) for the early Archean sources. This set of points is close to the LOMUVIPMAR composition (Figure 7b).

In the northern part of the Siberian paleocontinent, Putorana tholeiitic basalts show a steep trend with a slope of about 3.9 Ga (corrected for 250 Ma) and a series of points shifted to the right of this trend in lines whose slopes correspond to an age of about 0.38 Ga (corrected for 250 Ma) (Figure 7c). The old age means that for the protolith formation, the younger ones designate its late processing. The protolith age referred to the present time (4.15 Ga) exceeds the age interval of the early Archaean sources in unstable Asia (Figure 4).

This age interval is determined for sources of volcanic rocks from unstable Asia (South Tibet, Changbai, Jeungok, and Linju) and for sources of traps from the Indian and Siberian paleocontinents (Tapi rift and Noril'sk area, respectively).

The 2.81 Ga generation of a source for high-Mg volcanic rocks from South Tibet is the only recorded Late Archean mantle event on the background of the Early Archaean sources prevailing in the Indo-Asian collision zone (Figure 7).

In the Changbai volcanic field, there is a wide range of basic compositions (tholeiitic basalts, potassic trachybasalts, shoshonites, alkaline basalts) and differentiates (trachytes, alkaline trachytes, latites, commendites, and pantellerites). The basic volcanic rocks are subdivided into 4 groups (Figure 8a). Each group includes tholeiitic basalts, potassic trachybasalts, and shoshonites and only group 3—alkaline basalts. Group 1 that belongs mostly to the post-shield 2.0–0.9 Ma stage, shows a peculiar source related to the LOMUVIPMAR composition with an age estimate of 2.72 Ga. Groups 2 and 3 with elevated $^{207}Pb/^{204}Pb$ ratios yield age estimates about 2.93 Ga obtained for different photoliths. Finally, group 4 has an age estimate of about 1.99 Ga. A common component of groups 3 and 4 (Changbai common component) indicates their origin from a single source that denotes the lowest Pb isotope ratios ($^{206}Pb/^{204}Pb$ = 17.262, $^{207}Pb/^{204}Pb$ = 15.512). First, this source produced an array of about 2.93 Ga (group 3), then it generated another array of about 1.99 Ga (group 4).

Miocene volcanic rocks from the Linju volcanic field (Shandong Peninsula, China) designate a line, whose slope corresponds to an age of 2.57 ± 0.17 Ga [52]. Data points of lavas, erupted between 18.9 and 13.7 Ma, fall to the left of the LOMUVIPMAR composition and those of the younger lavas, erupted between 13.4 and 10.6 Ma, extend between LOMUVIPMAR and ELMUVIPMAR (Figure 8b).

In the Indian paleocontinent, an array of the Tapi Rift basalts has a slope corresponding to an age of 2.67 ± 0.18 Ga (corrected for 65 Ma). Groups of data points correspond to the LOMU and ELMU compositions and a single point to the HIMU value. In the Siberian paleocontinent, picrites and picrobasalts of the Gudchikha Formation (Noril'sk region) show a LOMU–ELMU array with a slope that corresponds to an age of about 2.36 Ga (corrected for 250 Ma) (Figure 8c).

## 2.4. Paleoproterozoic (2.0–1.8 Ga)

This age interval is obtained for sources of volcanic rocks from the unstable Asia (Changbai, Wudalianchi, Indochina, Hannuoba), Indian paleocontinent (Central Deccan, Kutch), and Siberian paleocontinent (Noril'sk region).

Sources of volcanic rocks from the Changbai volcano, dominated by early Archean processes, show the Paleoproterozoic event about 1.99 Ga as the minor array of group 4 that departed from the common component (Figure 8a). In contrast, sources of volcanic rocks from the Quaternary Wudalianchi volcanic field have only a Paleoproterozoic protolith. The 2.5 Ma trachyandesites of the Laoshantou flow and 2.0 Ma low-Mg trachyandesites of the old Gelaqiushan flow denote lithospheric

sources differentiated at 1.88 ± 0.06 Ga (MSWD = 0.67). These represent two portions of materials related to the common component ($^{206}$Pb/$^{204}$Pb = 17.55, $^{207}$Pb/$^{204}$Pb = 15.52), displayed in volcanic rocks of different areas of East Asia. Besides the Paleoproterozoic protolith, Wudalianchi volcanic rocks are indicative also for recording of the late Phanerozoic melting processes (Figure 9).

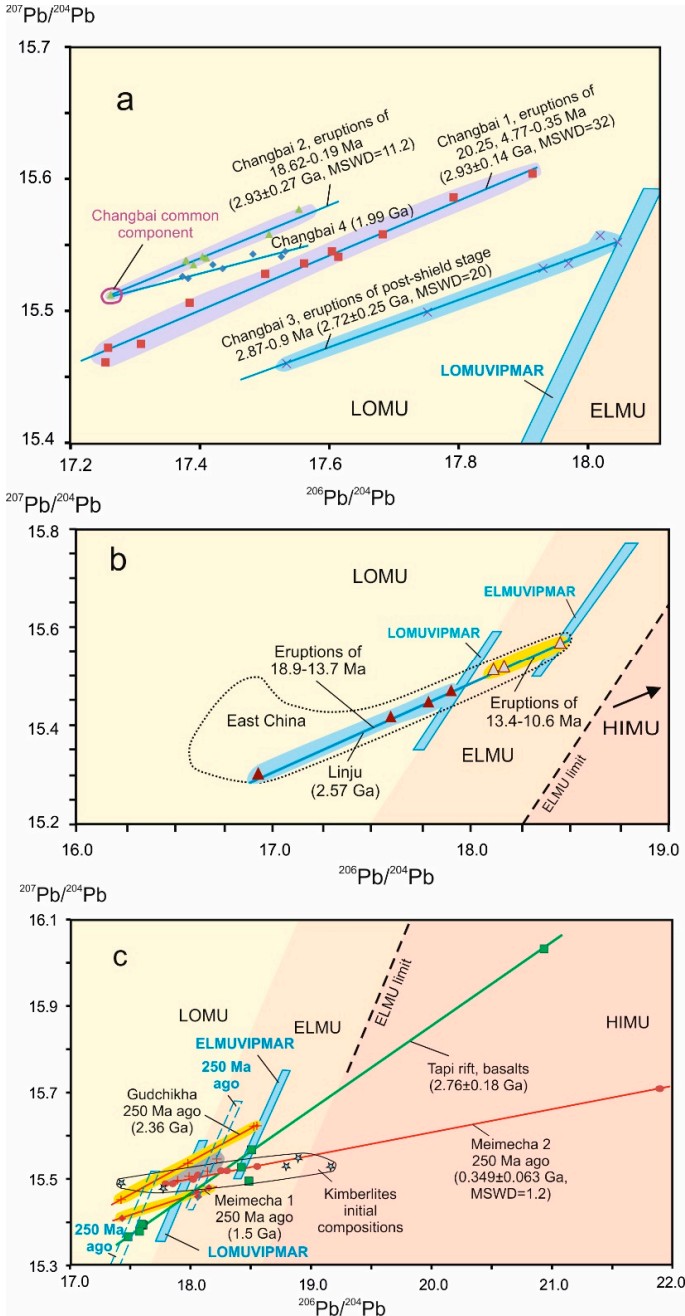

**Figure 8.** Diagrams of $^{207}$Pb/$^{204}$Pb versus $^{206}$Pb/$^{204}$Pb for basic rocks from volcanic fields of Northeast China ((**a**)—Changbai, (**b**)—Linju) and paleocontinents ((**c**)—Indian, Tapi Rift and Siberian, picrites and picrobasalts of the Gudchikha Formation, meimechite groups Meymecha 1 and 2). In diagrams (**a**,**b**), the LOMUVIPMAR and ELMUVIPMAR compositions, as well as LOMU–ELMU and ELMU limit discriminate lines are shown as in Figure 5. In diagram (**a**), the Changbai common component is defined at the least radiogenic Pb isotope composition ($^{206}$Pb/$^{204}$Pb = 17.26, $^{207}$Pb/$^{204}$Pb = 15.51). In diagram (**c**), initial Pb isotope compositions of kimberlites are plotted for comparison. The secondary isochrones were calculated using the Isoplot program [87]. Data are from [41,48,52,54,57,96–98].

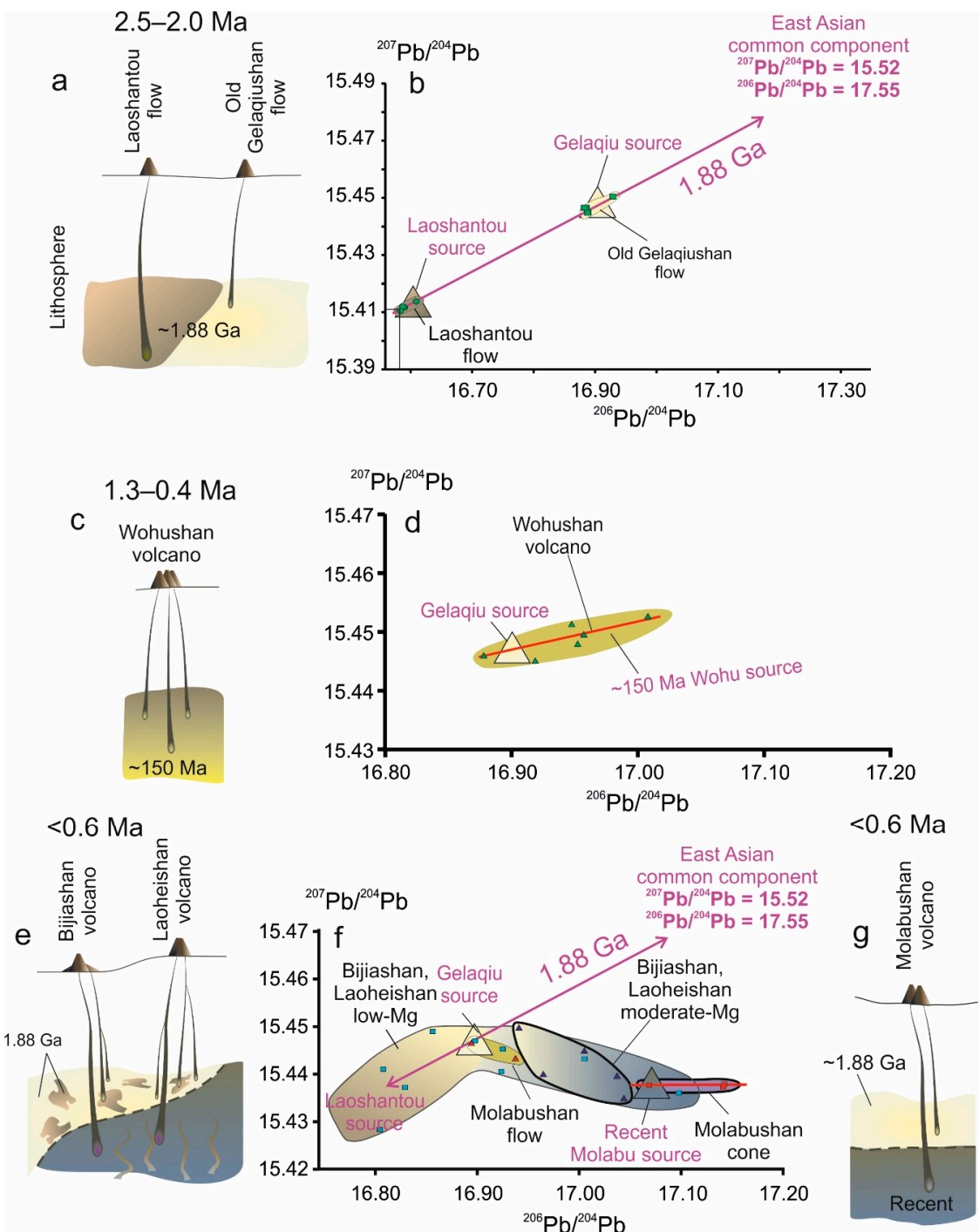

**Figure 9.** Illustrations of sources and diagrams $^{207}$Pb/$^{204}$Pb versus $^{206}$Pb/$^{204}$Pb of potassic lavas from the Wudalianchi volcanic field: (**a,b**)—Laoshantou and old Gelaqiushan flows, (**c,d**)—Wohushan volcano, (**e–g**)—Bijiashan, Laoheishan, and Molabushan volcanoes. In diagram (**b**), calculation of the secondary isochron 1.88 Ga was performed using the Isoplot program [87]. Modified after [40].

Similar to volcanic rocks of the Changbai and Wudalianchi volcanic fields, those of the Hannuoba volcanic field (North China) are also derived from sources of different ages. These volcanic rocks belong to the Miocene sequence that shows interbedded tholeiitic basalts (TB) and transitional, evolved, primitive basalts (TEPB). The rock groups reveal different trace-element and isotopic signatures [48,99]. The TB data points yield an array, whose slope corresponds to an age of about 1.81 Ga. The array extends from the LOMUVIPMAR composition to the LOMU field (Figure 10a).

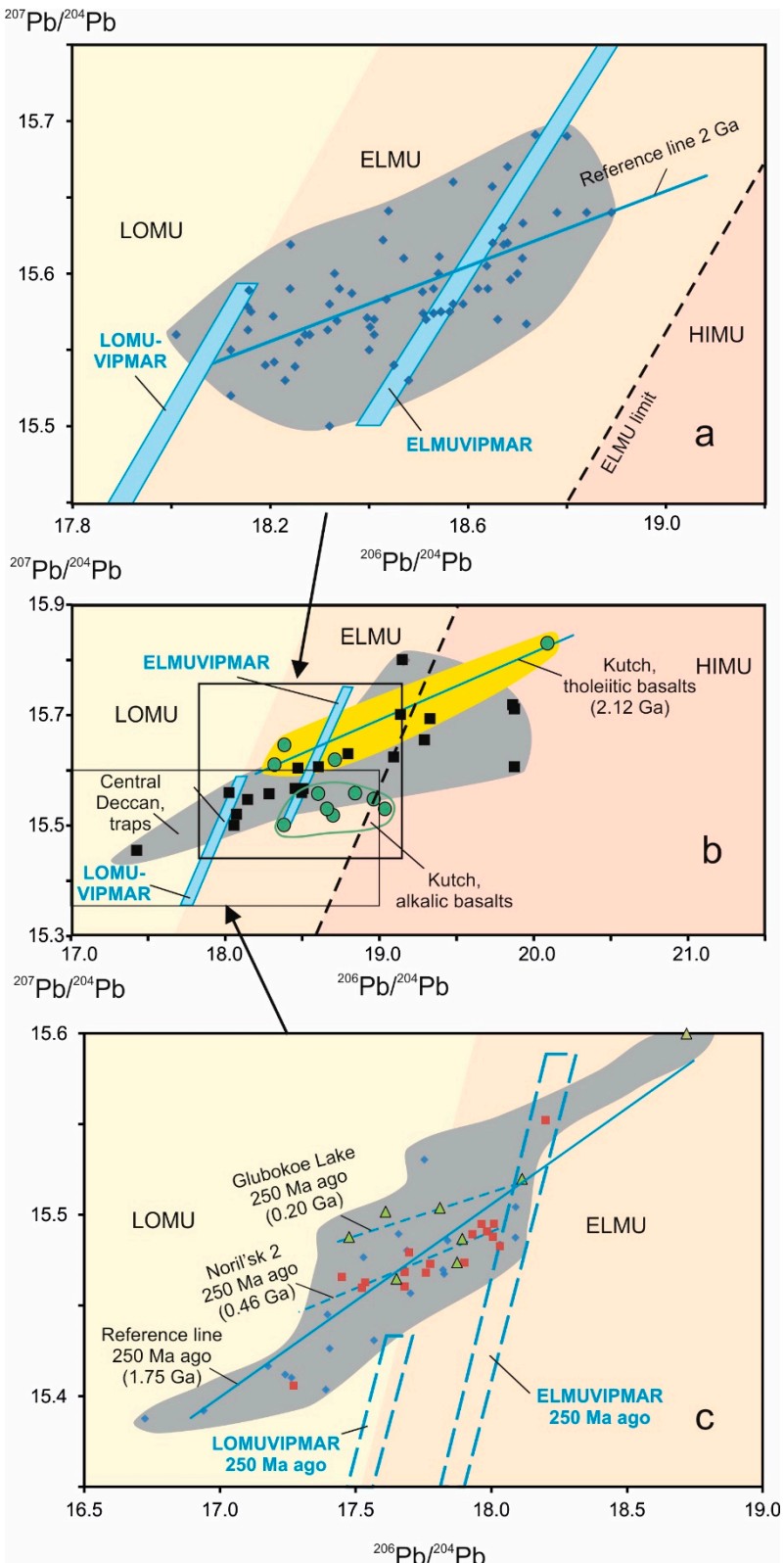

**Figure 10.** Diagrams of $^{207}Pb/^{204}Pb$ versus $^{206}Pb/^{204}Pb$ for volcanic and intrusive rocks from Indochina (**a**), Indian paleocontinent (**b**), and Siberian paleocontinent (Noril'sk area) (**c**). In diagrams (**a**,**b**), the LOMUVIPMAR and ELMUVIPMAR compositions, as well as LOMU–ELMU and ELMU limit discriminating lines are shown as in Figure 5. In diagram (**c**), data points of traps and viscous protomantle compositions correspond to the age of 250 Ma. Calculation of the secondary isochron 0.46 Ga was performed using the Isoplot program [87]. Data are from [41,94,98,100–108].

A trend of Cenozoic volcanic rocks from Indochina is compared with the 2 Ga OIB + MORB array. A whole data set but one point from this area is plotted within the ELMU field (Figure 11a).

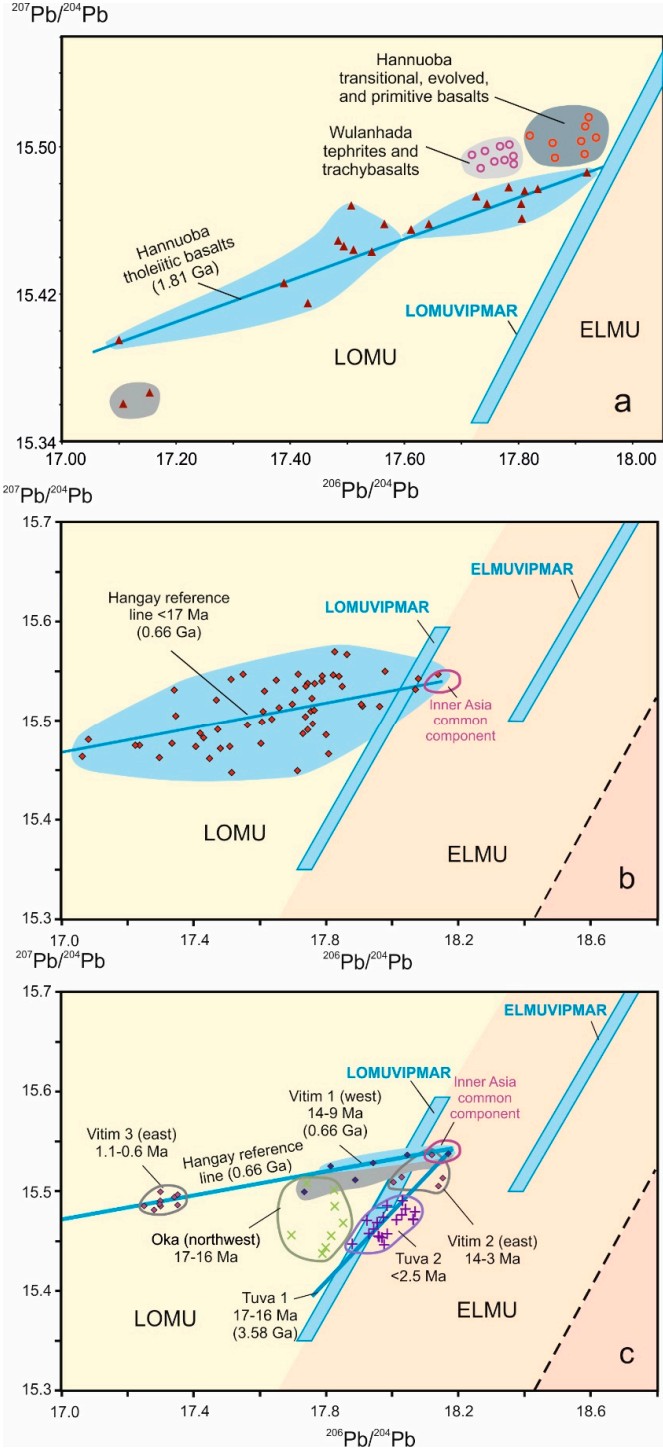

**Figure 11.** Diagrams of $^{207}Pb/^{204}Pb$ versus $^{206}Pb/^{204}Pb$ for volcanic rocks from the Hannuoba and Wulanhada volcanic fields (**a**), Hangay volcanic area of the southwestern part of the Baikal rift system (**b**), its northwestern part (Oka, Tuva 1, 2) and northeastern part (Vitim 1–3, Udokan 2) (**c**). The LOMUVIPMAR and ELMUVIPMAR compositions, as well as LOMU–ELMU and ELMU limit discriminating lines are shown as in Figure 5. In diagrams (**b**,**c**), designated is the Inner Asia common component of volcanic rocks. Data used are from [14,38,66,109–111] and unpublished data of authors.

In the Indian paleocontinent, scattered data points of the Cretaceous-Paleogene traps from the Deccan Plateau also fit into the 2 Ga OIB + MORB array (Figure 1). An age estimate about 2.12 Ga (corrected for 65 Ma) is obtained for tholeiitic basalts of the Kutch rift. Data points of alkaline basalts of the same area are shifted below the trend of tholeiitic basalts and are characterized by a gentle trend that indicates the generation of a younger source related to the ELMUVIPMAR composition. Pb isotope signatures of Central Deccan traps vary from the LOMU to HIMU values with the maximum concentration of data points in the ELMU field (Figure 10b).

In the Siberian paleocontinent, a scattered data set of traps from the Noril'sk region shows a similar general trend with a slope of about 2 Ga. The whole set but one point extends to the left of the ELMUVIPMAR composition corrected for 250 Ma (Figure 10c).

### 2.5. Neoproterozoic (0.7–0.6 Ga)

Neoproterozoic age estimates are obtained for sources of volcanic rocks from the Hangay volcanic area (Central Mongolia) and Vitim volcanic field. Similar ages are available for traps and kimberlites from the northern part of the Siberian paleocontinent.

Volcanic rocks from the Hangay volcanic area, erupted in the last 17 Ma, show an array of scattered data points mainly of LOMU values and slightly advanced into the ELMU field. The central approximating line has a slope corresponding to an age of about 0.66 Ga (Figure 11b). Most of the data points of volcanic rocks from the western part of the Vitim volcanic field, designated as Vitim 1, fall on this line. In terms of the Sm–Nd–Pb isotope systematics, the source of this rock group shows a common component typical of sources for volcanic rocks from Inner Asia [4] (Figure 11b,c).

In the northern part of the Siberian paleocontinent, meimechites (group Meimecha 2) show an extended LOMU–ELMU–HIMU array, similar to the steeper one of basalts from the Tapi rift. An age of the Meimecha 2 source (corrected for 250 Ma) is 0.349 Ga (0.599 Ga relative to the present). Several meimechite points yield a separate trend (Meimecha group 1) with a steeper slope corresponding to an age of about 1.5 Ga (corrected for 250 Ma) (Figure 8c). A sporadic distribution of data points of traps from the Noril'sk area may indicate the Paleoproterozoic mantle protolith affected by Neoproterozoic processing. Most data points of rocks from the Noril'sk 2 section belong to a trend with a slope of about 0.46 Ga (corrected for 250 Ma, 0.71 Ga relative to the present). Three points of traps from Glubokoe lake fall on this trend. Four other points fit a line with a slope of about 0.2 Ga (corrected for 250 Ma, 0.45 Ga relative to the present) (Figure 10c).

### 2.6. Late Phanerozoic (<0.25 Ga)

Late Phanerozoic ages are indicated for the Wohu and Molabu sources of the Wudalianchi volcanic field (Figure 9).

The Wohu source resulted from melting and differentiation of the Paleoproterozoic protolith. Lavas of the Wohushan volcano yielded an array originated through melting and differentiation of the Gelaqiu source material. Data points of low-Mg volcanic rocks from this volcano show an array with a slope of about 150 Ma. In contrast, the Molabu one, designated a new mantle component, had no relation to the Laoshantou or Gelaqiu sources. This source, displayed by the 0.6 Ma eruption of moderate-Mg tephrites, presents a Quaternary convecting mantle composition. Moderate- and low-Mg volcanic rocks of the last 0.6 Ma from the Bijiashan and Laoheishan volcanoes exhibit mixed materials of the Molabu, Laoshantou, and Gelaqiu sources (Figure 9e–g).

### 2.7. Indefinite Ages

Isometric distributions of data sets of volcanic rocks imply no direct age interpretations of Pb isotope data for the Oka, Tuva 2, Vitim 2, Vitim 3, Hannuoba TEPB, and Wulanhada sources. These may have only indirect age estimates.

The 17–16 Ma units of volcanic rocks from the Oka and East Tuva volcanic fields, located, respectively, within the Neoproterozoic Tuva-Mongolian microcontinent and Khamsara Caledonian

zone, are plotted near the LOMUVIPMAR composition. A data set of the former looks like an isometric field, a data set of the latter (Tuva 1) yields an array related to the Inner Asian common component with a slope that corresponds to an age of about 3.58 Ga. An isometric data set of volcanic rocks that erupted in the East Tuva volcanic field in the last 2.5 Ma (Tuva 2) falls on the central part of the 3.58 Ga array. In this case, the compact Tuva 2 data set has inherited the dated Early Archean Tuva 1 protolith (Figures 7b and 11c).

Both the Vitim 2 and Vitim 3 units, representing lava eruptions in the east of the Vitim volcanic field, respectively, 14–3 Ma ago and 1.1–0.6 Ma ago, have compact data sets. The Vitim 2 one falls to the right of the LOMUVIPMAR composition below the Vitim 1 trend. The Vitim 3 displaces along the line approximated the Hangay data set with an age of about 0.66 Ga (Figure 11c).

Data points of the Hannuoba TEPB group form a compact isometric field above the Hannuoba TB array near the LOMUVIPMAR composition. Those of the Wulanhada tephrites and trachybasalts also show an isometric data set (Figure 11a).

## 3. Discussion

The mantle of Asia likely preserves closed U–Pb isotopic signatures of its early consolidation at the shallow level (i.e., in the oldest lithospheric remnants) and at the deeper one (i.e., in the viscous lower mantle), while it may exhibit multiple events in intermediate depths (i.e., in the viscous upper mantle), subjected to fluid and magmatic processing. For that reason, we compare the LOMUVIPMAR and ELMUVIPMAR ages, on the one hand, with those of detrital zircons and protoliths of the accessible Earth and, on the other hand, we consider these ages in the context of the early Earth evolution as a space body. Further, we characterize age ranges of volcanic rock sources with the LOMU and ELMU signatures as stages of the upper mantle processing. In regional correlations of volcanic rock sources within Asia, we take into account the nature of their common components, while, in global correlations, we speculate about possible cause of the isolated evolution of the LOMU–ELMU sources in the mantle of Asia, in contrast to the HIMU component distributed worldwide.

### 3.1. Age Comparisons of the Viscous Protomantle with Hadean Detrital Zircons and Protoliths

The primary surface of the Earth has been destroyed by exogenous processes, so no evidence on the early Earth events had been known until the late 1980s, when the U–Pb (SHRIMP) age of 4.03–3.94 Ga was obtained for the Acasta gneisses (Canada) [112,113]. Later, the U–Pb age of about 4.2 Ga was measured for zircon xenocrysts from these gneisses using laser ablation combined with plasma mass spectrometry and a high-resolution ion probe measurement [114]. Pseudo-amphibolites, identified in the Nouvuwagittuk greenstone belt (Quebec, Canada), were interpreted as a material enriched with incompatible trace-elements shortly after the completion of primary planetary accretion. These rocks were characterized by isochronous $^{142}$Nd/$^{146}$Sm–$^{146}$Sm/$^{142}$Nd age of 4280 + 53/−81 Ma [115]. The Nulliac suite of the Saglek block (Labrador, Canada) was described as differentiated mantle material of 4.40 ± 0.03 Ga defined in the $^{176}$Lu/$^{177}$Hf system [116].

The famous location of Hadean detrital zircons with U–Pb ages up to 4.404 ± 0.008 Ga was described in Jack Hills (Western Australia) [117]. At present, detrital zircons were detected all around the world. In Asia, rocks of the Hadean–Archean boundary were studied in the Anshan area of the Sino-Korean Craton [118]. Detrital zircons of 3.981 ± 0.009 Ga were found in the Changdu block (North Kyangtang, Tibet). From a two-stage Hf-isotope model, their age estimates were obtained in the interval of 4.316–3.784 Ga. Similar detrital zircons of 4.1–4.0 Ga were found in the Himalayas [119]. Zircon xenocrysts containing cores older than 3.9 Ga with a model Lu/Hf ages up to 4.45 Ga were identified in Ordovician ignimbrites of the North Quinlin Belt (China) [120].

The Hadean mantle sources of Asia are compared with coeval volcanic rocks that have been identified in the Greenland and Baffin islands. For those from Isua Belt, the mantle differentiation time of 4.460 ± 0.115 Ga is estimated in the $^{146}$Sm–$^{142}$Nd system combined with the initial epsilon $^{143}$Nd value [121]. In the diagram of uranogenic leads (Figure 12), the Hadean volcanic rocks from

both Western Greenland and Baffin are plotted between the LOMUVIPMAR and ELMUVIPMAR compositions similar to those of the Hadean Udokan 1, Udokan 2, and Abaga-Dalenuor sources.

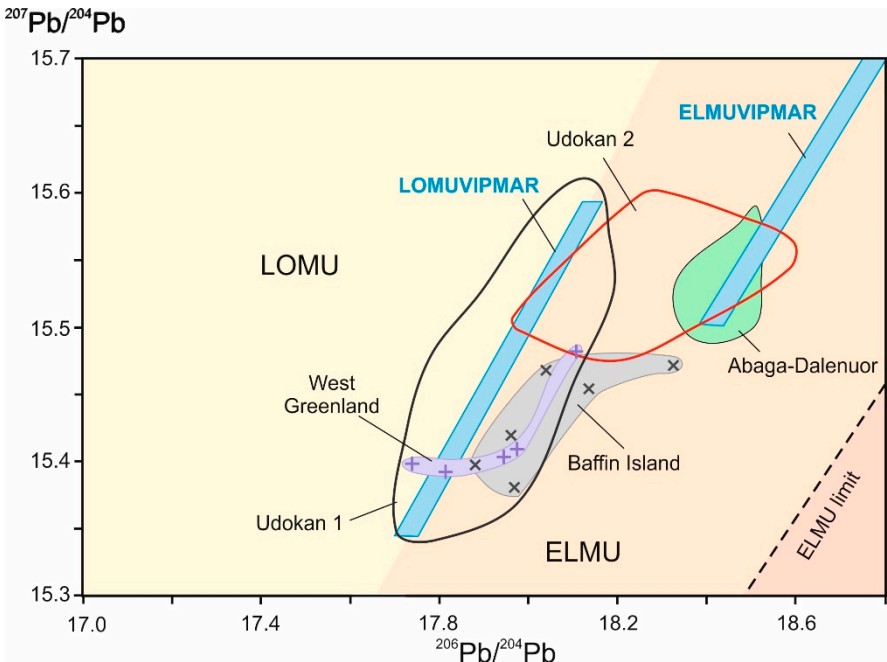

**Figure 12.** Diagram of $^{207}Pb/^{204}Pb$ versus $^{206}Pb/^{204}Pb$ for comparisons of Hadean volcanic rocks (Buffin Island and Western Greenland) and Hadean sources of volcanic rocks (Udokan 1, 2 and Abaga-Dalenuor). Data fields of volcanic rocks, the LOMUVIPMAR and ELMUVIPMAR compositions, as well as LOMU–ELMU and ELMU limit discriminate lines are shown as in Figure 5. Data are from [14,53,122].

The least radiogenic Pb isotopic compositions of the Hadean volcanic rocks correspond to those of the Udokan 1 source, while the most radiogenic ones have slightly reduced $^{207}Pb/^{204}Pb$ ratios relative to those of the Udokan 2 and Abaga-Dalenuor sources. This reflects the lower μ of erupted Hadean lavas relative to the μ of Hadean sources. The protolith generations of the Udokan 1 and Udokan 2 sources in the initial peripheral Earth's shell, along with crystallization of the Hadean volcanic rocks, yield Pb isotopic ratios decreased along the LOMUVIPMA and ELMUVIPMA geochrons due to the primary shallow differentiation of the protolith material 4.51 and 4.44 Ga ago. Pb isotopic ratios may decrease along the ELMUVIPMA geochron in the Abaga-Dalenuor source, relative to the Kunlun source, due to μ decreasing resulted from its shallow differentiation. Therefore, like a lower mantle protolith of the Gobi source, a protolith of the Kunlun source seems to belong to the viscous lower mantle, while, like a protolith of the lithospheric source Udokan 2, a protolith of the Abaga-Dalenuor source may originate in the shallow lithosphere.

### 3.2. Generation of the Viscous Protomantle

After the Sun formation at 4567.3 ± 0.16 Ma ago [83], ProtoEarth grew through the accretion of planetesimals. A core formed in 30 Ma [123]. As a result of a giant release of energy in the solar system, the melted Earth became a global magma ocean, in which siderophile and chalcophile elements were sequestered into the core [18,124,125]. In models of the growing Earth, each addition of metals from the outside during its accretion was balanced with the entire mantle with rejection into the core [125–128]. The exact timing of the giant impact that entailed the formation of the Moon is unknown. It is assumed that this event occurred no later than 60 Ma after the Sun formation.

Various models were proposed for primary accretion of the Earth. In one of these, the late veneer was limited to the time interval of 80–130 Ma after the formation of the Earth as a space body (i.e., in the

interval of 4.487–4.437 Ga) [35]. In another one, the initial planetary accretion was considered as an entry of dry enstatite chondrite-like material that was followed by the addition of bioelements (water) in the time interval of 4.37–4.20 Ga ago [129].

The study of the moon has shown that at the early stage of the Solar system evolution, the terrestrial planets experienced numerous impact events. An additional influx of meteorites after core formation is considered as a late veneer that is reflected in an excess of highly siderophile elements (Pt, Re, Os, Au) in the Earth's mantle [123,130]. At least part of the late veneer was added to the mantle source of Isua basalts before 3.8 Ga ago [131–135].

From the study of lunar rocks, a strong melting of the Moon (and the Earth) was constrained between 4.43 and 4.35 Ga. The interval was designated by the Lu–Hf model age of the crystallization residue of the urKREEP reservoir (enriched with incompatible elements K, REE, P) with a Lu/Hf chondrite ratio [136]. An extensive large-scale magmatic event about 4.37 Ga ago was dated in the Lu–Hf and Sm–Nd isotope systems of lunar samples as evidence on the later crystallization of the lunar magma ocean [137]. This moon event was confirmed by Pb–Pb age of 4.376 ± 0.018 Ga obtained for rocks of the urKREEP reservoir and ferruginous anorthosites (FAN) [138]. The age of the ELMUVIPMA reservoir corresponds to the beginning of the proposed strong melting in the Moon and Earth (Figure 13).

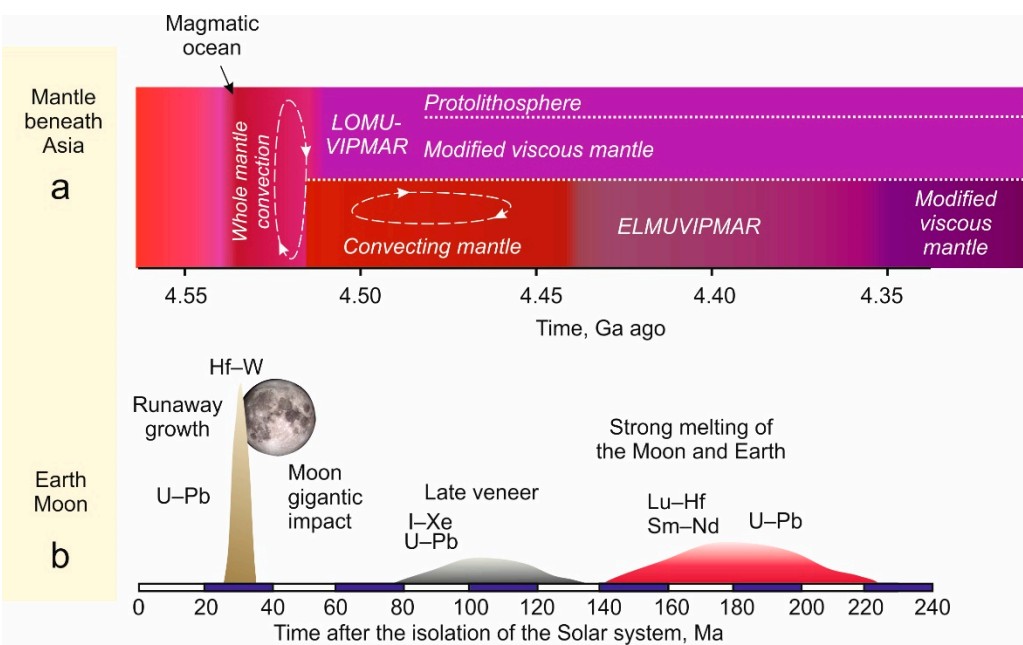

**Figure 13.** Generation of the LOMUVIPMAR at 4.51 Ga ago and later maintaining of low viscosity that provides the ELMUVIPMAR generation beneath Asia (**a**) in the context of the early Earth events (**b**). In panel (**b**), sequence of events is shown after [35,136–138].

### 3.3. Sources of the Oldest Lithosphere and Lower Mantle

Pb isotopic ages of sources for volcanic rocks of unstable Asia generally show no connection with ages of accessible crustal fragments. The lithospheric origin is assumed only for the oldest sources of volcanic rocks from the Udokan volcanic field, where the Udokan 1 and Udokan 2 sources yield, respectively, the LOMUVIPMAR composition and a mixture between the LOMUVIPMAR and ELMUVIPMAR components (Figure 5). A protolith of the Udokan 1 source is compared, on the one hand, with the composition of Hadean lavas (Figure 12), and, on the other hand, with a protolith of the Gobi source. The latter shows an array with the protolith age close to the one of LOMUVIPMAR (Figure 5).

Respectively, Cretaceous-Paleogene volcanic eruptions in Gobi are considered as a result of rising hot material from the lower mantle. This is also confirmed by a bilateral velocity decrease in the transition layer beneath the South Gobi [139]. No similar low-velocity anomaly is defined in the transition layer beneath the Udokan area. In contrast to the Gobi source region, the erupted LOMUVIPMAR materials of the Udokan 1 source and LOMUVIPMAR–ELMUVIPMAR mixture of the Udokan 2 source is likely derived from the reactivated protolithospheric mantle of the Siberian paleocontinent (Figure 1).

The ELMUVIPMA geochron slope, corresponding to an age of 4.44 Ga, is observed for data points of volcanic rocks from Kunlun, Abaga-Dalenuor, and Jeju (Figure 5). The Kunlun volcanic area corresponds to a low-velocity anomaly that extends at least to a depth of 700 km in the P-wave model [140]. Therefore, the erupted materials of the Kunlun melting anomaly could rise from the lower mantle. Similarly, Jeju Island volcanism is believed to have been caused by intrusion of a lower mantle material in the backside of the Northeastern Honshu island arc [49]. Although a deep root of the Jeju melting anomaly has not yet been confirmed by seismic tomography, the occurrence of the ELMUVIPMAR component may indicate its lower mantle origin.

A low-velocity anomaly is recorded in the upper mantle at depth of 250 km beneath the Abaga-Dalenuor volcanic field and adjacent areas of North China and Southwest Mongolia [139]. No velocity decrease is shown, however, by tomography models, e.g., [141] in the transition layer.

*3.4. Age Systematics of Mantle Sources*

Age ranges of mantle sources (Figure 4) denote 6 stages of the viscous mantle generation: (I) Hadean, (II) Early Archean, (III) Late Archean, (IV) Paleoproterozoic, (V) Neoproterozoic, and (VI) Late Phanerozoic (Figure 14).

I. The Hadean mantle stage (4.51–4.44 Ga) provides the isolation of LOMUVIPMAR and ELMUVIPMAR in the cooling magma ocean. Subsequently, these reservoirs supply the upper mantle with materials that show time-integrated LOMU and ELMU signatures.

II. The Early Archean mantle stage (4.0–3.7 Ga) is marked by both decreasing and increasing μ values relative to the LOMUVIPMAR and ELMUVIPMAR compositions. The ELMU signature is displayed in sources of melting anomalies in Tibet and Tien Shan (Lhasa, Songpan-Ganzi, Tuyun); the LOMU one is exhibited in melting anomalies of East Asia (Nushan, Jeungok) and Inner Asia (Tuva). In the north of the Siberian paleocontinent, the ELMUVIPMAR composition is defined in a source of Putorana traps.

In the hypothesis of long-lasted Earth accretion [124,130,142], the early Archean is considered as a time of the Late Heavy Bombardment (LHB) indicated by a frequency jump in the 4.1–3.8 Ga lunar-crater impacts. The LHB is suggested to result from an arrival of cosmic bodies from the periphery of the Solar System to the Earth and other terrestrial group planets [124]. A total mass of the late veneer material is estimated at 1.0–2.5% of the planet mass [142,143]. Early Earth's processes, recorded by ore leads in the Early Precambrian crust of the Gargan block and southern margin of the Siberian paleocontinent, yield parameters *T* of 4.31 and 3.82 Ga, respectively. The first closure of the U–Pb isotope system coincided with the end of the planetary accretion of the Earth, the second one with the end of its LHB [39].

III. The Late Archean mantle stage (2.9–2.6 Ga) is divided into phases a and b. During phase a, LOMU arrays are generated only in East Asia (Changbai, Jeungok). During phase b, short ELMU arrays are displayed in South Tibet and Tien Shan and a longer LOMU–ELMU one is generated in East Asia (Linju). These data sets of unstable Asia are complemented by a short LOMU–ELMU array of the Siberian paleocontinent (Gudchikha) and a longer LOMU–ELMU–HIMU array of the Indian paleocontinent (Tapi rift) (Figure 14b). This stage is notable by a significant increase in the rate of the continental crust grows (about 2.7 Ga ago) [144].

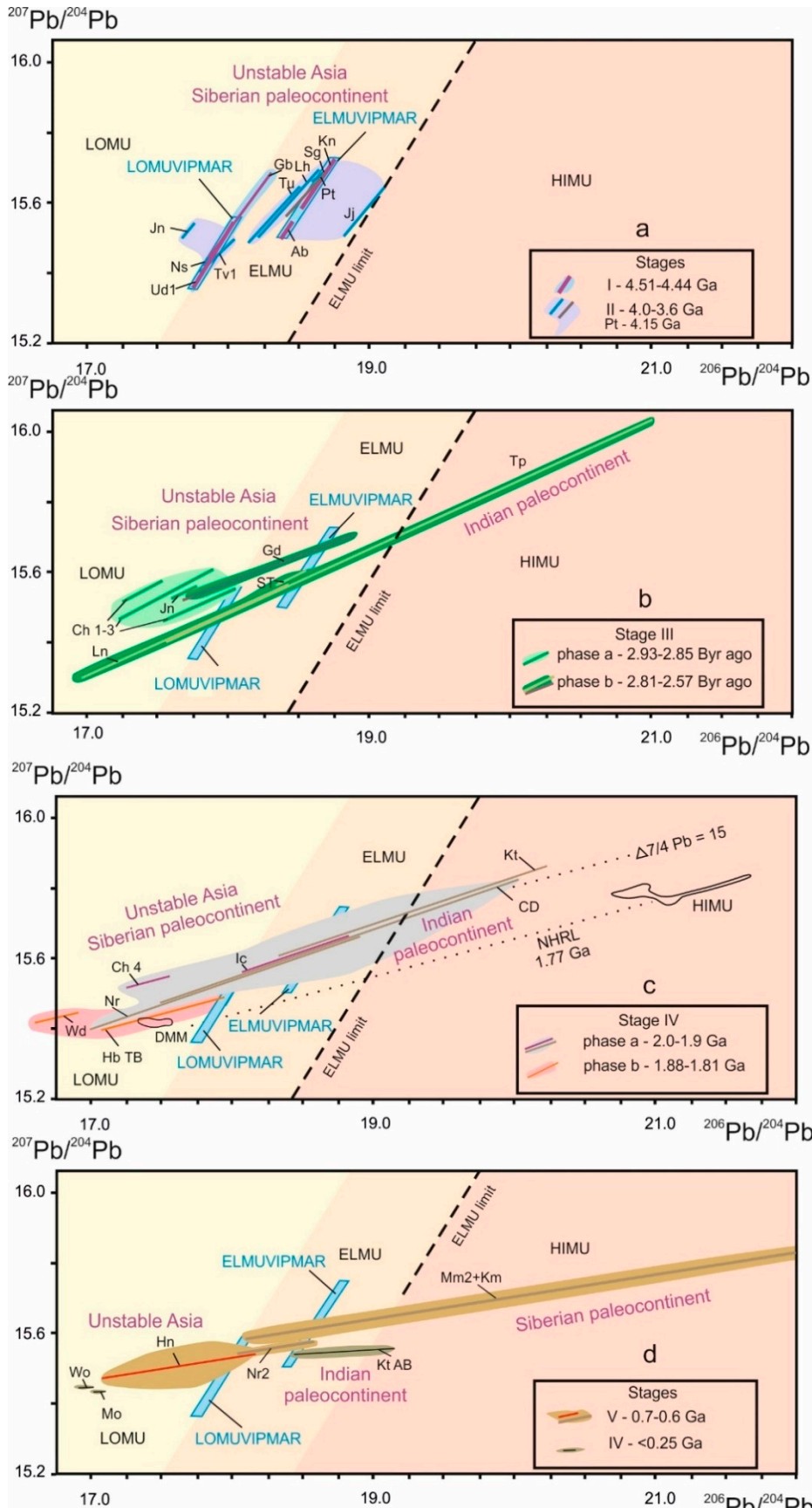

**Figure 14.** Mantle stages of Asia in summary diagrams of Pb isotopic data for sources of late Phanerozoic melting anomalies of unstable Asia and stable Siberian and Indian paleocontinents. Location of melting

anomalies is indicated in Figure 1. The LOMUVIPMAR and ELMUVIPMAR compositions and LOMU–ELMU and ELMU limit the discriminating lines are plotted as in Figure 5. Trends of 250 Ma rocks from the Siberian paleocontinent are recalculated to the present. (**a**)—the Early Mantle Epoch: the Hadean (I) and Early Archean (II) stages are indicated, respectively, by the LOMUVIPMAR–ELMUVIPMAR reservoirs and initial generation of LOMU–ELMU material (Udokan 2 source is not shown); (**b,c**)—the Middle Mantle Epoch: the Late Archean (III) stage is characterized by phases a (LOMU source trend, evolved from the LOMUVIPMAR) and b (wide LOMU–HIMU range in mantle sources of the Indian paleocontinent); the Paleoproterozoic (IV) stage is represented by phases a (rejuvenated wide LOMU–HIMU range in mantle sources of the Indian paleocontinent) and b (LOMU sources in the mantle of the unstable Asia with the generation of the maximal HIMU in the South Pacific and African domains); (**d**)—the Late Mantle Epoch: the Neoproterozoic (V) and late Phanerozoic (VI) stages are indicated, respectively, by LOMU convecting mantle sources in the unstable Asia and by the development of the HIMU meimechite–kimberlite trend in the northern part of the Siberian paleocontinent.

IV. The Paleoproterozoic mantle stage (2.0–1.8 Ga) is also displayed by phases a and b. During phase a, LOMU and ELMU arrays are generated in East Asia (Changbai 4 and Indochina, respectively) accompanied by a comparable array of the Siberian paleocontinent (Noril'sk traps) and longer LOMU–ELMU–HIMU ones of the Indian paleocontinent (Kutch rift and Central Deccan traps). Phase b is distinguished by activities of only LOMU sources displayed in East Asia (Hannuoba TB, Changbai 4, and Wudalianchi).

An Re–Os isotope age of 1.91 ± 0.22 Ga, obtained for sulfides from mantle xenoliths of the Hannuoba volcanic rocks [145], is supplemented by 2 Ga Re–Os model age estimates of mantle sources beneath the Siberian paleocontinent [146]. It is noteworthy that an Os of the Noril'sk complex is attributed to a plume source because of a significant enrichment of $^{186}$Os relative to other terrestrial and meteorite samples [147]. All these data may be indicative of the 2 Ga processes likely related to the core and deep mantle. The defined Paleoproterozoic time interval is characterized by the assembly of crustal blocks into the North China, Siberian, and other paleocontinents [148].

V. The Neoproterozoic mantle stage (0.7–0.6 Ga) is again associated only with the LOMU sources but is shifted from East to Inner Asia (Hangay and Vitim 1). Associated with this stage are the mantle events in the northern part of the Siberian paleocontinent, where the Meimecha 2 and kimberlite sources give the LOMU–ELMU–HIMU array with a slope of 0.599 Ga. Sources of the Putorana and Noril'sk areas are also strongly affected by the Neoproterozoic mantle processes. This mantle stage is associated with the remarkable structural reorganization of the Earth known as the breakup of the Rodinia supercontinent.

VI. The late Phanerozoic mantle stage is directly marked by sources of the convecting mantle of unstable Asia (Wohu and Molabu, Wudalianchi) and the Indian paleocontinent (Kutch AB). It is noteworthy also that the entire spectrum of mantle sources of both unstable Asia and stable paleocontinents, identified in the mantle stages I–V, is reactivated at this very stage.

The whole sequence of sources in unstable Asia (Figure 14) yields three major periods of the mantle evolution: Early Mantle Epoch (Hadean—Early Archean), Middle Mantle Epoch (Late Archean—Paleoproterozoic), and Late Mantle Epoch (Neoproterozoic—Late Phanerozoic). In the Early Mantle Epoch, the Hadean LOMUVIPMAR and ELMUVIPMAR protoliths yield the LOMU and ELMU viscous upper mantle generated until the Early Archean. Over time, the role of an ELMU material decreases, and the role of a LOMU material increases. In the Middle Mantle Epoch, the Late Archean LOMU–ELMU sequence is supplemented by the Paleoproterozoic one with the opposite (ELMU–LOMU) order. In the Late Mantle Epoch, the Neoproterozoic and Late Phanerozoic sources are composed exclusively from a LOMU material.

In the viscous upper mantle of the northern stable part of the Siberian paleocontinent, ELMU is generated in the second half of the Early Mantle Epoch (Putorana), ELMU and LOMU are distributed in

the Middle Mantle Epoch (Gudchikha, Noril'sk), and LOMU, ELMU, and HIMU are widespread in the Late Mantle Epoch (Meimecha 2, kimberlites). In contrast, no sources are detected in the viscous upper mantle of the stable Indian paleocontinent in the Early Mantle Epoch, the long LOMU–ELMU–HIMU arrays are recognized in the Middle Mantle Epoch (Tapi Rift, Kutch TB, Central Deccan), and a short ELMU array, slightly penetrated into the HIMU field, is recorded in the Late Mantle Epoch (Kutch AB). So, on the one hand, sources of the stable continents differ from those of unstable Asia by occurrence of the HIMU signature, on the other hand, the LOMU–ELMU and HIMU sources of the Indian and Siberian paleocontinents show opposite change in the Earth's history. In the Indian paleocontinent, the HIMU signature appears in the Middle Mantle Epoch and degrades in the Late one, in the Siberian paleocontinent, it is absent at the end of the Early Mantle Epoch and at the beginning of the Middle one, but develops at the end of the Middle Mantle Epoch (about 2 Ga ago) and culminates later.

Only few LIPs occur in the Early Mantle Epoch (Figure 15). In the Middle Mantle Epoch, their role increases (Bushveld, Circum-Super, etc.). The Late Mantle Epoch corresponds to frequent LIPs (Siberian, Emeishan, Ontong Java, Dean, Ethiopian, etc.). Among the four robust age matches between southern Siberia and northern Laurentia 1870, 1750, 1350, and 720 Ma ago [149], the extreme values indicate, respectively, the end of the Middle Mantle Epoch and the beginning of the late one. It is noteworthy also that the Abitibi, Murphy, Vell, Perseverens, Kambalda, Aleko, Munro, and Hauper komatiites erupted at the beginning of the Middle Mantle Epoch and the Thompson, Gilmore, and Raglan ones at its end, although the Barberton, Commondale, Ruth Vell, Weltevreden komatiites, which occurred between 3.4 and 3.2 Ga ago [150], fall between the Early and Middle Mantle Epochs. The Gargona komatiites correspond to the beginning of the latest geodynamic stage in Asia about 90 Ma ago [139].

### 3.5. Permanent versus Changing Common Components

Common components are different in mantle sources of Inner and East Asia. In Inner Asia, the only component is specified in sources of different ages to the right of LOMUVIPMAR. This component is defined in the 3.58 Ga Tuva 1 source and in the 0.66 Ga Hangay and Vitim 1 ones. In East Asia, common components change their compositions from stage to stage in accordance with a temporal ELMU to LOMU transition. The oldest common component in the Jeju source is due to a Hadean homogenization and subsequent differentiation of the ELMUVIPMAR composition (stages I–II, 4.44–4.0 Ga) (Figure 5b). This common component is followed by the Jeungok one (stages II–III, 3.94–2.85 Ga) (Figure 6c), then by the Changbai and East Asian common components (stages III–IV, 2.93–1.99 Ga ago and stages IV, 1.88 Ga ago, respectively) (Figures 8a and 9), and, finally, by the Wohu and Molabu sources of the Wudalianchi volcanic field (stages IV–VI, 1.88–0.15 Ga ago and stage VI of the latest convective mantle, respectively) (Figure 9). Thus, the young Wohu and Molabu sources are located at the end of the general temporal trend that stretches from the common ELMUVIPMAR component to LOMU values (Figure 16). It is noteworthy that potassic Wudalianchi volcanic rocks show no trace-element signatures of oceanic basalts, but present specific features of the primordial mantle [40].

A rapid change from the 1.99 Ga Changbai common component to the 1.88 Ga East Asian one is compared with the Wohu to Molabu change of sources in the past 0.15 Ga. A similarity between the Changbai–East Asian and Wohu–Molabu reorganizations may indicate a significant effect of the Quaternary Molabu source reactivation comparable to the effect of Paleoproterozoic homogenization and differentiation of the East Asian mantle material resulted in a long 1.88 Ga array.

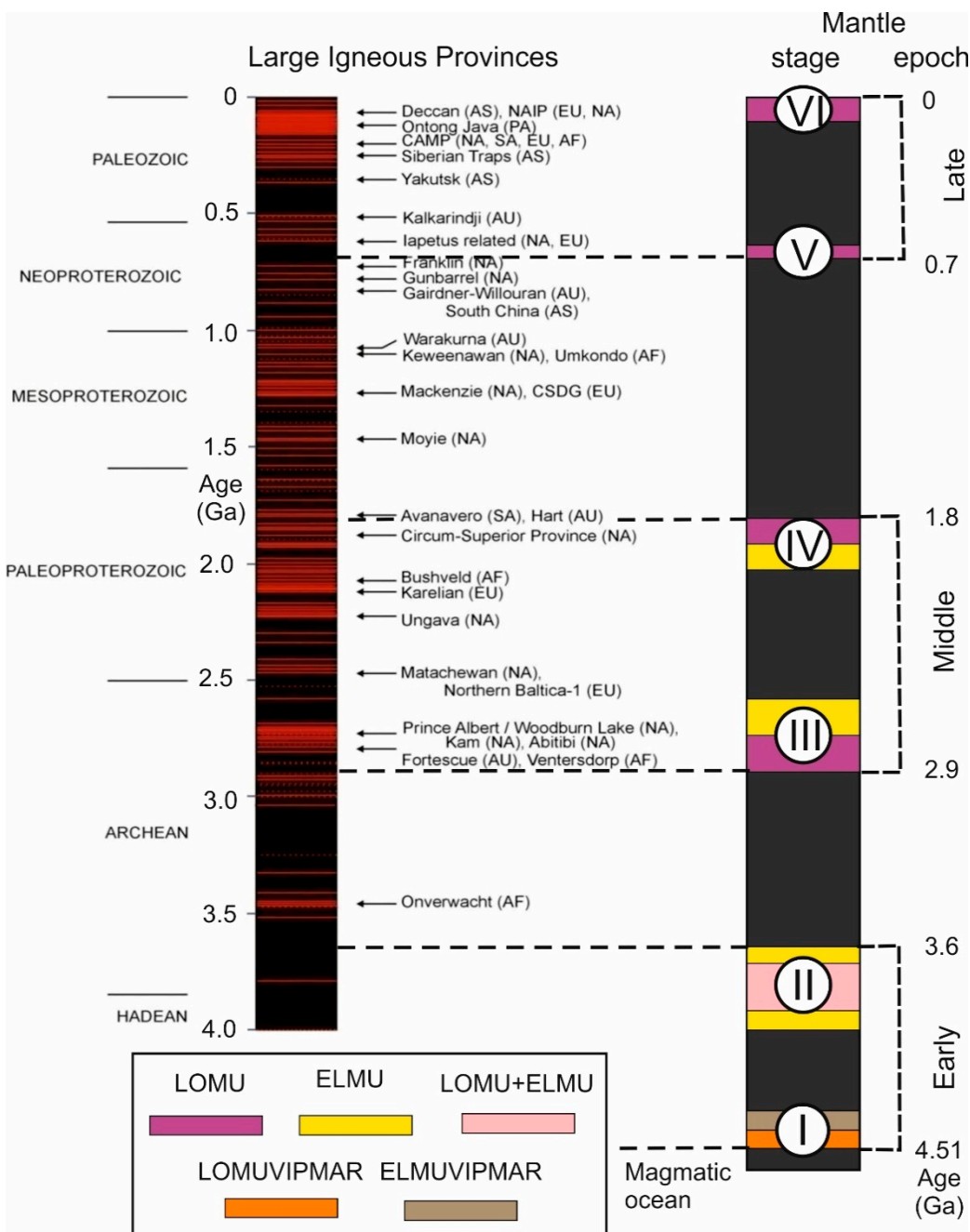

**Figure 15.** Comparison of the age mantle systematics, defined in sources of volcanic rocks from unstable Asia, with LIPs recorded world-wide. The LIP barcode is shown after [151].

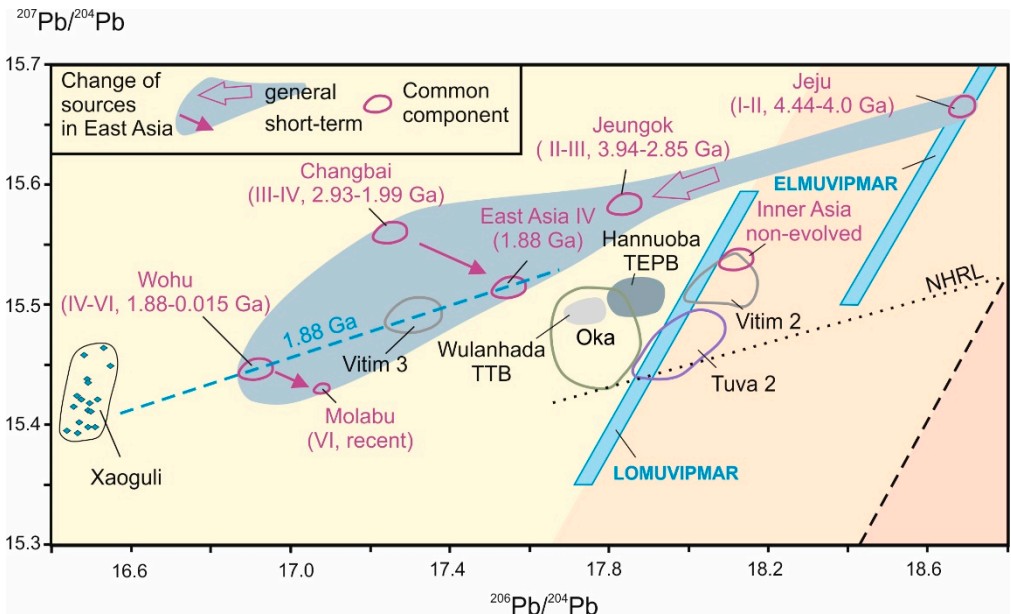

**Figure 16.** Diagram of $^{207}Pb/^{204}Pb$ versus $^{206}Pb/^{204}Pb$ for common components and isometric data fields of volcanic rocks from Inner and East Asia. The LOMUVIPMAR and ELMUVIPMAR compositions, as well as the LOMU–ELMU and ELMU limit dividing lines are shown as in Figure 5. Shown are the common components of the Figures 5b, 6c, 8a, 9 and 11b,c and isometric data fields of the Figure 11a,c. A data field of the Xiaoguli volcano is plotted after [65].

From experimental data [152], U shows greater incompatibility than Pb that suggests the generation of low U/Pb ratios in the depleted mantle. The changing common components of the mantle beneath East Asia may reflect stepwise depletion in a source region starting from the ELMUVIPMAR composition, in contrast to the LOMUVIPMAR-related common component, that was displayed permanently in the mantle source region beneath Inner Asia.

Beside common components of volcanic rocks, isometric data fields of volcanic rocks are also plotted in Figure 16 as indicators of homogenized sources (in terms of their μ signatures). In East Asia, this source type is exhibited by ultrapotassic rocks from the Xiaoguli volcano erupted in the Quaternary Wudalianchi zone [153]. The ultrapotassic data set is plotted at the end of the East Asia–Wohu array of common components. Therefore, it is likely resulted from an advanced depletion of the mantle material in a source region about 1.88 Ga ago.

Other isometric data fields (Vitim 2, Tuva 2, Oka, Hannuoba TEPB, Wulanhada) occur to the left and below the non-evolved common component of Inner Asia (Figure 16). These compositions may present mantle materials that have been slightly differentiated from the LOMUVIPMAR composition with subsequent U–Pb homogenization. A specific composition is shown only by a compact isometric (basanite–tephriphonolite) Vitim 3 set that is significantly shifted to the left of the non-evolved LOMUVIPMAR common component. Moreover, it is plotted on the ELMUVIPMAR-related trend of common components. If the Vitim 3 source has been generated about 0.66 Ga ago (Section 2.7, Figure 11c), it can be considered as a time-intermediate (stage V) link between the East Asian common component, generated at the stage IV (1.88 Ga ago), and the Wuhu one, strongly displaced along the isochron line by the stage VI (0.15 Ga ago) (Figure 16).

Unlike LOMU–ELMU sources of volcanic rocks from the Japan-Baikal geodynamic corridor, those of the Tibet-Indochina geodynamic belt have only ELMU signatures. The ELMUVIPMAR to LOMU common-component evolution in the mantle (characteristic of the East Asian mantle source region) shows no evidence on such advanced change in the ELMU mantle of the Tibet-Indochina geodynamic belt. These kinds of processes is comparable with those in a mantle source region of Inner Asia, where slightly differentiated LOMU material predominates. We infer that varied μ signatures indicate (1)

the weak depletion of the LOMU mantle in Inner Asia, (2) the weak depletion of the ELMU mantle in the Tibet-Indochina geodynamic belt, and (3) the pronounced time-integrated depletion of the ELMU mantle in East Asia, combined with the LOMU mantle components.

### 3.6. LOMU–ELMU versus HIMU Components in the Global Geodynamic Context

In the presented review of Pb isotopic signatures of volcanic rocks, only LOMU and ELMU mantle sources have been recognized in unstable Asia and the HIMU component has been identified in sources of stable regions of the Indian and Siberian paleocontinents (Figure 17). In discussions about the origin of the HIMU component, various options for increasing $^{238}U/^{204}Pb$ (μ) in deep sources has been proposed due to anomalous uranium enrichment or lead depletion of a subducted oceanic material [77]. From integrated isotopic studies of the core formation, increase of μ values was explained by sequestration of Pb from the mantle into the core as a chalcophile element. The importance of this process was emphasized especially at the early stage of the Earth's formation [17,18,35]. From the pronounced HIMU array, it was inferred that Pb was separated from the mantle to the core by sulfides about 2 Ga ago [11,18].

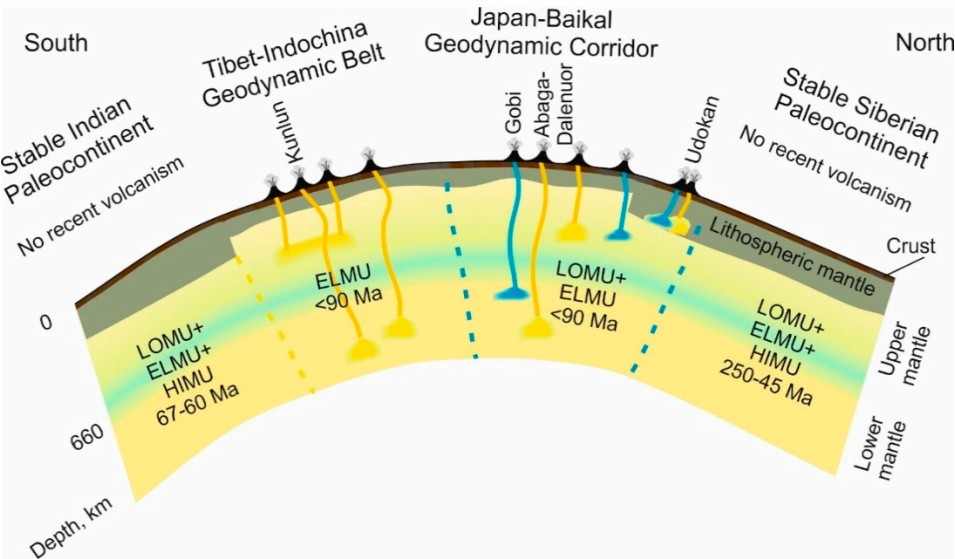

**Figure 17.** North-south change from ELMU to ELMU–LOMU mantle sources for volcanic rocks of the latest geodynamic stage in unstable Asia and transitions to LOMU–ELMU–HIMU sources of volcanic rocks in stable regions of the Siberian and Indian paleocontinents.

While connecting the HIMU component with the Pb sulfide sequestration, it is noteworthy that sources of the Early Mantle Epoch of unstable Asia do not show compositions beyond ELMU, i.e., these were not affected by the core. Moreover, the LOMUVIPMAR and ELMUVIPMAR and derivative LOMU and ELMU sources of the Early Mantle Epoch could survive in the mantle of unstable Asia only in a case of no influence on it of the Middle Mantle Epoch processes. Indeed, in the Middle and Late Mantle Epochs, the role of the ELMU material in mantle sources of East Asia notably decreased, as the role of the LOMU material increased.

Due to sulfide sequestration of Pb from the mantle to the core, a HIMU material is displayed in the mantle of the Tapi rift in the Indian paleocontinent in the Late Archean, at the beginning of the Middle Epoch. In the Paleoproterozoic HIMU peak, its generation is also provided by this effect in Central Deccan and Kutch rift. In the northern part of the Siberian paleocontinent, the ELMU and LOMU signatures are generated in the mantle of the Early and Middle Mantle Epochs and are supplemented by the HIMU values only in the meimechite and kimberlite sources generated in the Late Mantle Epoch (about 0.6 Ga ago). A north-south change in the mantle domains of unstable Asia is established in the

context of a global heterogeneity of the Earth's mantle associated with the influence of the core on the processes occurred in the mantle.

It has been suggested that the core of the early Earth was liquid and that a solid core formed later [154–156]. Consequently, the LOMU–ELMU limitation of the viscous mantle sources beneath Asia in the Early Mantle Epoch may indicate a harmonious relationship between the core and mantle that prevented generations of any global mantle heterogeneities.

An instability factor of the solid inner core was expressed in its globally differentiated effect on the mantle, indicated by the HIMU component generation in source regions of the Middle Mantle Epoch. A strong resonance of the HIMU component could occur in the South Pacific and African low-velocity lower mantle domains, in contrast to the HIMU-free sources of unstable Asia that correspond to the high-velocity lower mantle domain (Figure 18). Inherited generation of the LOMU–ELMU viscous mantle beneath unstable Asia could occur only in a case if the Asian mantle domain exhibited a conservative skeleton, the lower mantle of which was subjected to a neutral or downwelling mode and never was experienced an upwelling regime throughout the Earth's history.

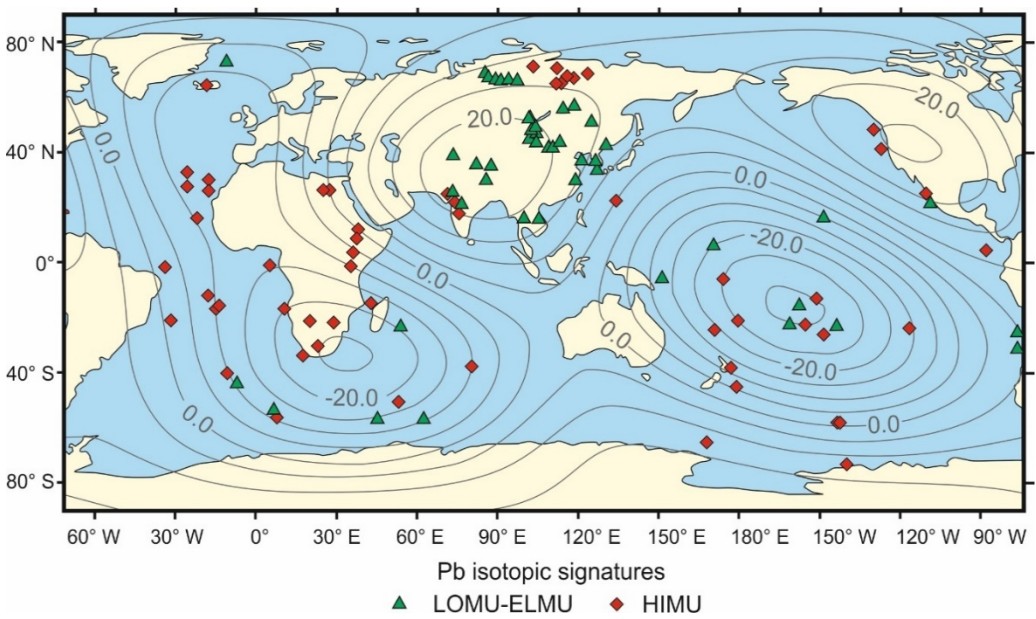

**Figure 18.** Distribution of melting anomalies with LOMU–ELMU signatures in the Asian high-velocity domain and those with HIMU signatures in the South Pacific and African low-velocity domains. Isolines of the P-wave velocities averaged over the entire lower mantle are shown after [1]. Indicated in Asia are the melting anomalies with the LOMU–ELMU and HIMU signatures of Figure 1 and in other regions of the world are those adopted from the reviews by Jackson et al. [5] and Homrighausen et al. [6].

Meanwhile, the asynchronous HIMU generation in the Indian paleocontinent about 2.74 Ga ago and in the Siberian paleocontinent about 0.6 Ga ago (Figure 8c), suggests a change in geodynamics from the Middle to the Late Mantle Epoch with a transfer of the core influence from the mantle of the southern part of the African domain to the mantle of Arctic. The core-mantle dynamics, displayed mostly in low-velocity lower mantle and HIMU resonance in mantle sources of the Southern Hemisphere, develops in the Arctic mantle in the late Phanerozoic. Consequences of the Late Phanerozoic reorganization of the Earth are breakup of the Arctic paleocontinent, a LIP generation, and opening of Arctic Ocean [157–162].

The core imbalance in the Middle and Late Mantle Epochs could generate differentiated internal forces in the Earth's interior that resulted in its complicated evolution. Motions of lithospheric plates and tectono-stratigraphic terranes in the upper shell of the accessible Earth likely adapted to the deep core–mantle dynamics.

## 4. Conclusions

The developed Pb-isotopic age systematics of the mantle sources for Asian volcanic rocks is based on the theoretical predictions of the Earth's evolution with varying viscosity in mantle layers. This process is displayed in open and closed evolution of the U–Pb isotope system of mantle protoliths. Viscosity decrease and transition to vigorous convection result in Pb isotope homogenization of a source region. Conversely, viscosity increase stops convection and leads to differentiation of a material that produces time-integrated Pb arrays of data points marked ages of sources for erupted volcanic rocks.

Mantle-derived volcanic rocks from Asia show $^{207}$Pb–$^{206}$Pb geochrons referred to reservoirs of protolithosphere and viscous lower protomantle generated due to cooling of the Hadean magma ocean about 4.51 Ga ago (LOMUVIPMAR) and about 4.44 Ga ago (ELMUVIPMAR). Besides, volcanic rocks yield secondary $^{207}$Pb–$^{206}$Pb isochrons and errorchrons that mark the generation of LOMU and ELMU viscous upper mantle portions in the Early Archean (4.0–3.7 Ga), Late Archean (2.9–2.6 Ga), Paleoproterozoic (2.0–1.8 Ga), Neoproterozoic (0.7–0.6 Ga), and Late Phanerozoic (<0.25 Ga).

From analysis of common components of volcanic rocks from East Asia, it is inferred that $^{207}$Pb/$^{204}$Pb and $^{206}$Pb/$^{204}$Pb ratios decreased stepwise in the upper mantle from the common Hadean component of the Jeju source, corresponding to ELMUVIPMAR, to the LOMU common late Phanerozoic components of the Wohu and Molabu sources. This trend is consistent with the expected time-integrated depletion of the mantle. In volcanic rocks from Inner Asia, the non-evolved common component, close to the LOMUVIPMAR composition, is determined. Conservative upper mantle processes result in weak depletion of the LOMU mantle. A comparable slightly evolved source material is defined in volcanic rocks from the Tibet-Indochina geodynamic belt, but, in contrast to Inner Asia, it is related to the ELMUVIPMAR composition processed into the ELMU mantle.

The cooperative occurrence of LOMU and ELMU sources in the Japan-Baikal geodynamic corridor and ELMU sources in the Tibet-Indochina geodynamic belt of unstable Asia change to mantle sources of stable regions of the Siberian and Indian paleocontinents with LOMU–ELMU–HIMU signatures. In mantle sources of the Late Phanerozoic melting anomalies, the HIMU component shows worldwide distribution. The spatial correspondence of the LOMU–ELMU sources to the Asian high-velocity lower mantle domain contrasts with the HIMU signature of sources spatially associated with the South Pacific and African low-velocity lower mantle domains.

Since the HIMU signature in mantle sources may result from Pb sulfide sequestration to the core about 2 Ga ago [13,18], it is assumed that these processes were due to an imbalance of the solid inner core designated the Middle Mantle Epoch. The core effects were focused in the South Pacific and African low-velocity mantle domains without their displaying in the Asian high-velocity mantle domain. Asynchronous generation of a HIMU material in mantle sources of the Indian and Siberian paleocontinents (in the Middle and Late Mantle Epochs, respectively) may indicate a relative decrease of the imbalanced core effect in the mantle of the southern African domain and its increase in the Arctic mantle.

**Author Contributions:** Conceptualization, S.R.; Methodology, Software, I.C., T.Y., and E.S.; Formal Analysis, S.R., I.C., and T.Y.; Investigation, S.R., I.C.; Resources, S.R.; Data Curation, S.R., T.Y., E.S.; Writing—Original Draft Preparation, S.R., T.Y., I.C., E.S.; Writing—Review & Editing, S.R., I.C., and T.Y.; Visualization, S.R., I.C., and T.Y.; Supervision, S.R. and I.C.; Project Administration, I.C.; Funding Acquisition, I.C., S.R. All authors have read and agreed to the published version of the manuscript.

**Funding:** This work has been funded by the RSF grant 18-77-10027 (I. Chuvashova, S. Rasskazov).

**Acknowledgments:** We are grateful to reviewers for useful comments. Analytical determinations of Pb isotopic compositions were performed at the Massachusetts Institute of Technology using a Sector 54 mass spectrometer, at the Institute of the Earth's Crust SB RAS using a Finnigan MAT 262 mass spectrometer, and at the Institute of Geochemistry SB RAS using a Neptun Plus multy-collector inductively coupled plasma mass spectrometer (MC-ICP-MS).

**Conflicts of Interest:** The authors declare no conflict of interest.

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
