# Peer review of "Mantle Evolution of Asia Inferred from Pb Isotopic Signatures of Sources for Late Phanerozoic Volcanic Rocks"

_minerals, doi:10.3390/min10090739_

Round 1

Reviewer 1 Report

The authors propose a different model of Pb isotope evolution for the proto-mantle based on evaluation of Pb-isotope data from volcanic rocks originating from different tectonic setting in Asia. They have made a tremendous work of analyzing and evaluating Pb isotopic data from different case studies. I think their model deserves to be published since it provides an innovative interpretation in the mantle heterogeneity in Asia region which is supported by their data. The problem I noticed is that the references in the text are not cited in a sequence, but are cited randomly, i.e. the first reference is [24]. Moreover references are not usually cited in the abstract except if it is absolutely necessary.

Author Response

Response to Reviewer 1 Comments

Point 1: The authors propose a different model of Pb isotope evolution for the proto-mantle based on evaluation of Pb-isotope data from volcanic rocks originating from different tectonic setting in Asia. They have made a tremendous work of analyzing and evaluating Pb isotopic data from different case studies. I think their model deserves to be published since it provides an innovative interpretation in the mantle heterogeneity in Asia region which is supported by their data. The problem I noticed is that the references in the text are not cited in a sequence, but are cited randomly, i.e. the first reference is [24]. Moreover references are not usually cited in the abstract except if it is absolutely necessary.

Response 1: The references have been cited in a sequence according to requirements of Minerals and have been omitted from the abstract.

Reviewer 2 Report

Major comment:

This paper proposes a new chemo-geodynamics model of the Earth’s mantle based on the systematic investigations of Pb-Pb isotopic systematics of continental volcanic rocks. It looks to me that the data compilation and interpretation are thorough and support the major conclusions of this paper except for the definition of LOMUVIPMAR and ELMUVIPMAR. LOMUVIPMAR and ELMUVIPMAR are empirical geochrons proposed by this study. While these geochrons play critical roles in this paper, it seems unclear to me how to define them. I would like to see how the authors obtained them (e.g., slope, ranges of 207Pb/204Pb) in the paper.

Introduction:

Sections 1-4 looks all “Introduction.”  How about changing the sub-section numbers and titles in the “1. Introduction” section as:

“1. Introduction” to “1.1. Backgrounds”

“2. Isotopic age assessment...” to “1.2 Isotopic age assessment...”

“3. Previous interpretations of...” to “1.3. Previous interpretations of...”

“4. Strategy of analysis of...” to “1.4 Strategy of analysis of...”

Style of references:

The style of references should follow the MINERALS author guidelines. The reference number does not appear in numerical order in the main text. 

Typo:

P8: “Another are” should be “Another is.”

Author Response

Response to Reviewer 2 Comments

Point 1: Major comment:

This paper proposes a new chemo-geodynamics model of the Earth’s mantle based on the systematic investigations of Pb-Pb isotopic systematics of continental volcanic rocks. It looks to me that the data compilation and interpretation are thorough and support the major conclusions of this paper except for the definition of LOMUVIPMAR and ELMUVIPMAR. LOMUVIPMAR and ELMUVIPMAR are empirical geochrons proposed by this study. While these geochrons play critical roles in this paper, it seems unclear to me how to define them. I would like to see how the authors obtained them (e.g., slope, ranges of 207Pb/204Pb) in the paper.

Response 1: Because of data point scattering, typical of Hadean sources, the presented LOMUVIPMA  and ELMUVIPMA ages have been estimated approximately, without using the Isoplot program [87]. To make this clear, the text of the manuscript has been slightly modified (highlighted by yellow color).

Page 7. The diagrams of uranogenic Pb isotope ratios (Figures. 2, 3) indicate a locus that corresponds to a transition from the magma ocean of the molten Earth to a Low m Viscous Protomantle (LOMUVIPMA), starting from the primordial lead of the Canyon Diablo meteorite and passing to the right of the meteorite geochron with a slope of 0.625034. A slope of the LOMUVIPMA geochron (0.600827), corresponding to an age of about 4.51 Ga, and a range of a Low m Viscous Protomantle Reservoir (LOMUVIPMAR) are derived from Pb isotope compositions of volcanic rocks from the northwestern part of the Udokan volcanic field.

A slope of the ELMUVIPMA geochron (0.572614), corresponding to an age of about 4.44 Ga, and a range of an Elevated μ Viscous Protomantle Reservoir (ELMUVIPMAR) are obtained from Pb isotope ratios of volcanic rocks of the Kunlun and Abaga-Dalenuor volcanic areas.

 In Figure 2, all geochrons are focused on the primordial lead isotope ratios determined in a troilite from the Canyon Diablo meteorite (Nanton): 207Pb/204Pb = 10.307094, 206Pb/204Pb = 9.305875 [84].

Page 10. In Figure 5a, the LOMUVIPMA geochron with a slope of about 4.51 Ga symmetrically approximates a scattered set of data points of volcanic rocks from the northwestern part of the Udokan volcanic field designated as the Udokan 1 unit. A reason to refer this line to a geochron is its orientation at the primordial lead composition of the meteorite Canyon Diablo. Respectively, the LOMUVIPMAR composition is determined by a range of Pb isotope ratios of the Udokan 1 volcanic rocks (207Pb/204Pb from 15.35 to 15.60 and 206Pb/204Pb from 17.7 to 18.2).

Page 11. The ELMUVIPMA locus is obtained for basalts, alkaline basalts, and basanites erupted in the Kunlun volcanic area in the last 18 Ma and also for volcanic rocks of the same compositions erupted in the Abaga-Dalenuor volcanic field in the last 15.4 Ma. The ELMUVIPMAR composition is constrained as a composite range of Kunlun and Abaga-Dalenuor volcanic rocks (207Pb/204Pb from 15.5 to 15.75 and 206Pb/204Pb from 18.4 to 18.8) (Figure 5b).

Point 2: Introduction:

Sections 1-4 looks all “Introduction.”  How about changing the sub-section numbers and titles in the “1. Introduction” section as:

“1. Introduction” to “1.1. Backgrounds”

“2. Isotopic age assessment...” to “1.2 Isotopic age assessment...”

“3. Previous interpretations of...” to “1.3. Previous interpretations of...”

“4. Strategy of analysis of...” to “1.4 Strategy of analysis of...”

Response 2: This suggestion has been accepted. In the new version, introduction is clearly separated from results and discussion.

Point 3: Style of references:

The style of references should follow the MINERALS author guidelines. The reference number does not appear in numerical order in the main text. 

Response 3: The references have been cited in a sequence according to requirements of Minerals and have been omitted from the abstract.

Point 4:

P8: “Another are” should be “Another is.”

Response 4: In the page 8, we have changed the text: “Some secondary isochrons are diverged from LOMUVIPMAR towards a low 238U/204Pb (LOMU) part of the Pb–Pb isotope diagram. Others are located around ELMUVIPMAR and referred to Elevated µ (ELMU) compositions”.
